# On Statistical Bias In Active Learning: How and When to Fix It

**Sebastian Farquhar**[†*]**, Yarin Gal**[†]**, Tom Rainforth**[‡*]
University of Oxford, [†]OATML, Department of Computer Science; [‡]Department of Statistics

## Abstract

Active learning is a powerful tool when labelling data is expensive, but it introduces a bias because the training data no longer follows the population distribution. We formalize this bias and investigate the situations in which it can be harmful and sometimes even helpful. We further introduce novel corrective weights to remove bias when doing so is beneficial. Through this, our work not only provides a useful mechanism that can improve the active learning approach, but also an explanation of the empirical successes of various existing approaches which ignore this bias. In particular, we show that this bias can be actively helpful when training overparameterized models—like neural networks—with relatively little data.

## 1 Introduction

In modern machine learning, unlabelled data can be plentiful while labelling requires scarce resources and expert attention, for example in medical imaging or scientific experimentation. A promising solution to this is active learning—picking the most informative datapoints to label that will hopefully let the model be trained in the most sample-efficient way possible (Atlas et al., 1990; Settles, 2010).

However, active learning has a complication. By picking the most informative labels, the acquired dataset is *not* drawn from the population distribution. This sampling bias, noted by e.g., MacKay (1992); Dasgupta & Hsu (2008), is worrying: key results in machine learning depend on the training data being identically and independently distributed (i.i.d.) samples from the population distribution. For example, we train neural networks by minimizing a Monte Carlo estimator of the population risk. If training data are actively sampled, that estimator is biased and we optimize the wrong objective. The possibility of bias in active learning has been considered by e.g., Beygelzimer et al. (2009); Chu et al. (2011); Ganti & Gray (2012), but the full problem is not well understood. In particular, methods that remove active learning bias have been restricted to special cases, so it has been impossible to even establish whether removing active learning bias is helpful or harmful in typical situations.

To this end, we show how to remove the bias introduced by active learning with minimal changes to existing active learning methods. As a stepping stone, we build a Plain Unbiased Risk Estimator, $\tilde{R}_{\text{PURE}}$, which applies a corrective weighting to actively sampled datapoints in pool-based active learning. Our Levelled Unbiased Risk Estimator, $\tilde{R}_{\text{LURE}}$, builds on this and has lower variance and additional desirable finite-sample properties. We prove that both estimators are unbiased and consistent for arbitrary functions, and characterize their variance.

Interestingly, we find—both theoretically and empirically—that our bias corrections can simultaneously also reduce the variance of the estimator, with these gains becoming larger for more effective acquisition strategies. We show that, in turn, these combined benefits can sometimes lead to significant improvements for both model *evaluation* and *training*. The benefits are most pronounced in underparameterized models where each datapoint affects the learned function globally. For example, in linear regression adopting our weighting allows better estimates of the parameters with less data.

On the other hand, in cases where the model is overparameterized and datapoints mostly affect the learned function locally—like deep neural networks—we find that correcting active learning bias can be ineffective or even harmful during model *training*. Namely, even though our corrections typically produce strictly superior statistical estimators, we find that the bias from standard active learning can actually be helpful by providing a regularising effect that aids *generalization*. Through this, our work explains the known empirical successes of existing active learning approaches for training deep models (Gal et al., 2017b; Shen et al., 2018), despite these ignoring the bias this induces.

---

*Equal contribution. Corresponding author `sebastian.farquhar@cs.ox.ac.uk`

To summarize, our main contributions are:

1. We offer a formalization of the problem of statistical bias in active learning.
2. We introduce active learning risk estimators, $\tilde{R}_{\text{PURE}}$ and $\tilde{R}_{\text{LURE}}$, and prove both are unbiased, consistent, and with variance that can be less than the naive (biased) estimator.
3. Using these, we show that active learning bias can hurt in underparameterized cases like linear regression but help in overparameterized cases like neural networks and explain why.

## 2 BIAS IN ACTIVE LEARNING

We begin by characterizing the bias introduced by active learning. In supervised learning, generally, we aim to find a decision rule $f_\theta$ corresponding to inputs, $\mathbf{x}$, and outputs, $\mathbf{y}$, drawn from a population data distribution $p_{\text{data}}(\mathbf{x}, \mathbf{y})$ which, given a loss function $\mathcal{L}(\mathbf{y}, f_\theta(\mathbf{x}))$, minimizes the population risk:

$$r = \mathbb{E}_{\mathbf{x}, \mathbf{y} \sim p_{\text{data}}} \left[ \mathcal{L}(\mathbf{y}, f_\theta(\mathbf{x})) \right].$$

The population risk cannot be found exactly, so instead we consider the empirical distribution for some dataset of $N$ points drawn from the population. This gives the empirical risk: an unbiased and consistent estimator of $r$ when the data are drawn i.i.d from $p_{\text{data}}$ and are independent of $\theta$,

$$\hat{R} = \frac{1}{N} \sum_{n=1}^{N} \mathcal{L}(\mathbf{y}_n, f_\theta(\mathbf{x}_n)).$$

In pool-based active learning (Lewis & Gale, 1994; Settles, 2010), we begin with a large unlabelled dataset, known as the pool dataset $\mathcal{D}_{\text{pool}} \equiv \{\mathbf{x}_n | 1 \leq n \leq N\}$, and sequentially pick the most useful points for which to acquire labels. The lack of most labels means we cannot evaluate $\hat{R}$ directly, so we use the *sub-sample empirical risk* evaluated using the $M$ actively sampled labelled points:

$$\tilde{R} = \frac{1}{M} \sum_{m=1}^{M} \mathcal{L}(\mathbf{y}_m, f_\theta(\mathbf{x}_m)). \tag{1}$$

Though almost all active learning research uses this estimator (see Appendix D), it is not an unbiased estimator of either $\hat{R}$ or $r$ when the $M$ points are actively sampled. Under active—i.e. non–uniform—sampling the $M$ datapoints are not drawn from the population distribution, resulting in a bias which we formally characterize in §4. See Appendix A for a more general overview of active learning.

Note an important distinction between what we will call "statistical bias" and "overfitting bias." The bias from active learning above is a statistical bias in the sense that using $\tilde{R}$ biases our estimation of $r$, regardless of $\theta$. As such, optimizing $\theta$ with respect to $\tilde{R}$ induces bias into our optimization of $\theta$. In turn, this breaks any consistency guarantees for our learning process: if we keep $M/N$ fixed, take $M \to \infty$, and optimize for $\theta$, we no longer get the optimal $\theta$ that minimizes $r$. Almost all work on active learning for neural networks currently ignores the issue of statistical bias.

However, even without this statistical bias, indeed even if we use $\hat{R}$ directly, the training process itself also creates an overfitting bias: evaluating the risk using training data induces a dependency between the data and $\theta$. This is why we usually evaluate the risk on held-out test data when doing model selection. Dealing with overfitting bias is beyond the scope of our work as this would equate to solving the problem of generalization. The small amount of prior work which does consider statistical bias in active learning entirely ignores this overfitting bias without commenting on it.

In §3-6, we focus on statistical bias in active learning, so that we can produce estimators that are valid and consistent, and let us optimize the intended objective, not so they can miraculously close the train–test gap. From a more formal perspective, our results all assume that $\theta$ is chosen independently of the training data; an assumption that is almost always (implicitly) made in the literature. This ensures our estimators form valid objectives, but also has important implications that are typically overlooked. We return to this in §7, examining the interaction between statistical and overfitting bias.

## 3 UNBIASED ACTIVE LEARNING: $\tilde{R}_{\text{PURE}}$ AND $\tilde{R}_{\text{LURE}}$

We now show how to unbiasedly estimate the risk in the form of a weighted expectation over actively sampled data points. We denote the set of actively sampled points $\mathcal{D}_{\text{train}} \equiv \{(\mathbf{x}_m, \mathbf{y}_m) | 1 \leq m \leq M\}$, where $\forall m : \mathbf{x}_m \in \mathcal{D}_{\text{pool}}$. We begin by building a "plain" unbiased risk estimator, $\tilde{R}_{\text{PURE}}$, as a stepping stone—its construction is quite natural in that each term is individually an unbiased estimator of the risk. We then use it to construct a "levelled" unbiased risk estimator, $\tilde{R}_{\text{LURE}}$, which is an unbiased

and consistent estimator of the population risk just like $\tilde{R}_{\text{PURE}}$, but which reweights individual terms to produce lower variance and resolve some pathologies of the first approach. Both estimators are easy to implement and have trivial compute/memory requirements.

## 3.1 $\tilde{R}_{\text{PURE}}$: PLAIN UNBIASED RISK ESTIMATOR

For our estimators, we introduce an active sampling proposal distribution over *indices* rather than the more typical distribution over datapoints. This simplifies our proofs, but the two are algorithmically equivalent for pool-based active learning because of the one–to–one relationship between datapoints and indices. We define the probability mass for each index being the next to be sampled, once $\mathcal{D}_{\text{train}}$ contains $m - 1$ points, as $q(i_m; i_{1:m-1}, \mathcal{D}_{\text{pool}})$. Because we are learning actively, the proposal distribution depends on the indices sampled so far ($i_{1:m-1}$) and the available data ($\mathcal{D}_{\text{pool}}$, note though that it does *not* depend on the labels of unsampled points). The only requirement on this proposal distribution for our theoretical results is that it must place non-zero probability on all of the training data: anything else necessarily introduces bias. Considerations for the acquisition proposal distribution are discussed further in §3.3. We first present the estimator before proving the main results:

$$\tilde{R}_{\text{PURE}} \equiv \frac{1}{M} \sum\nolimits_{m=1}^{M} a_m; \quad \text{where} \quad a_m \equiv w_m \mathcal{L}_{i_m} + \frac{1}{N} \sum\nolimits_{t=1}^{m-1} \mathcal{L}_{i_t}, \tag{2}$$

where the loss at a point $\mathcal{L}_{i_m} \equiv \mathcal{L}(\mathbf{y}_{i_m}, f_\theta(\mathbf{x}_{i_m}))$, the weights $w_m \equiv 1/Nq(i_m; i_{1:m-1}, \mathcal{D}_{\text{pool}})$ and $i_m \sim q(i_m; i_{1:m-1}, \mathcal{D}_{\text{pool}})$. For practical implementation, $\tilde{R}_{\text{PURE}}$ can further be written in the following more computationally friendly form that avoids a double summation:

$$\tilde{R}_{\text{PURE}} = \frac{1}{M} \sum_{m=1}^{M} \left( \frac{1}{Nq(i_m; i_{1:m-1}, \mathcal{D}_{\text{pool}})} + \frac{M - m}{N} \right) \mathcal{L}_{i_m}. \tag{3}$$

However, we focus on the first form for our analysis because $a_m$ in (2) has some beneficial properties not shared by the weighting factors in (3). In particular, in Appendix B.1 we prove that:

**Lemma 1.** *The individual terms $a_m$ of $\tilde{R}_{PURE}$ are unbiased estimators of the risk: $\mathbb{E}[a_m] = r$.*

The motivation for the construction of $a_m$ directly originates from constructing an estimator where Lemma 1 holds while only making use of the observed losses $\mathcal{L}_{i_1}, \ldots, \mathcal{L}_{i_m}$, taking care with the fact that each new proposal distribution does not have support over points that have already been acquired. Except for trivial problems, $a_m$ is essentially unique in this regard; naive importance sampling (i.e. $\frac{1}{M} \sum_{m=1}^{M} w_m \mathcal{L}_{i_m}$) does not lead to an unbiased, or even consistent, estimator. However, the overall estimator $\tilde{R}_{\text{PURE}}$ is not the only unbiased estimator of the risk, as we discuss in §3.2.

We can now characterize the behaviour of $\tilde{R}_{\text{PURE}}$ as follows (see Appendix B.2 for proof)

**Theorem 1.** *$\tilde{R}_{PURE}$ as defined above has the properties:*

$$\mathbb{E}\left[\tilde{R}_{PURE}\right] = r,$$

$$\text{Var}\left[\tilde{R}_{PURE}\right] = \frac{\text{Var}\left[\mathcal{L}(\mathbf{y}, f_\theta(\mathbf{x}))\right]}{N} + \frac{1}{M^2} \sum_{m=1}^{M} \mathbb{E}_{\mathcal{D}_{\text{pool}}, i_{1:m-1}} \left[\text{Var}\left[w_m \mathcal{L}_{i_m} | i_{1:m-1}, \mathcal{D}_{\text{pool}}\right]\right]. \tag{4}$$

**Remark 1.** *The first term of (4) is the variance of the loss on the whole pool, while the second term accounts for the variance originating from the active sampling itself given the pool. This second term is $O(N/M)$ times larger and so will generally dominate in practice as typically $M \ll N$.*

Armed with Theorem 1, we can prove the consistency of $\tilde{R}_{\text{PURE}}$ under standard assumptions: $\tilde{R}_{\text{PURE}}$ converges in expectation (i.e. its mean squared error tends to zero) as $M \to \infty$ under the assumptions that $N > M$, $\mathcal{L}(\mathbf{y}, f_\theta(\mathbf{x}))$ is integrable, and $q(i_m; i_{1:m-1}, \mathcal{D}_{\text{pool}})$ is a valid proposal in the sense that it puts non-zero mass on each unlabelled datapoint. Formally, as proved in Appendix B.3,

**Theorem 2.** *Let $\alpha = N/M$ and assume that $\alpha > 1$. If $\mathbb{E}\left[\mathcal{L}(\mathbf{y}, f_\theta(\mathbf{x}))^2\right] < \infty$ and*

$$\exists \beta > 0 : \min_{n \in \{1:N \setminus i_{1:m-1}\}} q(i_m = n; i_{1:m-1}, \mathcal{D}_{\text{pool}}) \geq \beta/N \quad \forall N \in \mathbb{Z}^+, m \leq N,$$

*then $\tilde{R}_{PURE}$ converges in its $L^2$ norm to $r$ as $M \to \infty$, i.e., $\lim_{M \to \infty} \mathbb{E}\left[(\tilde{R}_{PURE} - r)^2\right] = 0$.*

## 3.2 $\tilde{R}_{\text{LURE}}$: Levelled Unbiased Risk Estimator

$\tilde{R}_{\text{PURE}}$ is natural in that each term is an unbiased estimator of $r$. However, this creates surprising behaviour given the sequential structure of active learning. For example, with a uniform proposal distribution—equivalent to not actively learning—points sampled earlier are more highly weighted than later ones and $\tilde{R}_{\text{PURE}} \neq \tilde{R}$. Specifically, a uniform proposal, $q(i_m; i_{1:m-1}, \mathcal{D}_{\text{pool}}) = \frac{1}{N-m+1}$, gives a weight on each sampled point of $1 + \frac{M-2m+1}{N} \neq 1$. Similarly, as $M \rightarrow N$ (such that we use the full pool) the weights also fail to become uniform: setting $M = N$ gives a weight for each point of $1 + \frac{M-2m+1}{N} \neq 1$. $\tilde{R}_{\text{LURE}}$ fixes this. We first quote the estimator before proving key results:

$$\tilde{R}_{\text{LURE}} \equiv \frac{1}{M} \sum_{m=1}^{M} v_m \mathcal{L}_{i_m}; \quad v_m \equiv 1 + \frac{N-M}{N-m} \left( \frac{1}{(N-m+1)\, q(i_m; i_{1:m-1}, \mathcal{D}_{\text{pool}})} - 1 \right). \quad (5)$$

This estimator ensures that the expected value of the weight, $v_m$, does not depend on the *position* it was sampled in, but only on the probability with which it was sampled. That is, $\mathbb{E}[v_m] = 1$ for all $m$, $M$, $N$, and $q(i_m; i_{1:m-1}; \mathcal{D}_{\text{pool}})$. As a consequence the variance is generally lower. Moreover, we resolve the finite-sample behaviour shown by $\tilde{R}_{\text{PURE}}$. The weights become more even as $M$ increases for a given $N$, and when $M = N$, each $v_m = 1$ such that $\tilde{R}_{\text{LURE}} = \tilde{R} = \hat{R}$. Additionally, if the proposal is uniform, all weights are always exactly 1 such that $\tilde{R}_{\text{LURE}} = \tilde{R}$.

To derive $\tilde{R}_{\text{LURE}}$ note that each $a_m$ estimates $r$ without bias so for any normalized linear combination:

$$\mathbb{E}\left[ \frac{\sum_{m=1}^{M} c_m a_m}{\sum_{m=1}^{M} c_m} \right] = r,$$

provided that the $c_m$ are constant with respect to the data and sampled indices (they can depend on $M$, $N$, and $m$). In Appendix B.4 we show that the choice of :

$$c_m = \frac{N(N-M)}{(N-m)(N-m+1)}$$

produces the $v_m$ from (5) and in turn that these $v_m$ have the desired property $\mathbb{E}[v_m] = 1$, $\forall m \in \{1, \ldots, M\}$. We note also that $\sum_{m=1}^{M} c_m = M$, such that $\tilde{R}_{\text{LURE}} = \frac{1}{M} \sum_{m=1}^{M} c_m a_m$. We further characterise the variance and unbiasedness of $\tilde{R}_{\text{LURE}}$ as follows (see Appendix B.5 for proof)

**Theorem 3.** $\tilde{R}_{LURE}$ *as defined above has the following properties:*

$$\mathbb{E}\left[ \tilde{R}_{LURE} \right] = r,$$

$$\text{Var}\left[ \tilde{R}_{LURE} \right] = \frac{\text{Var}\left[ \mathcal{L}(\mathbf{y}, f_\theta(\mathbf{x})) \right]}{N} + \frac{1}{M^2} \sum_{m=1}^{M} c_m^2 \mathbb{E}_{\mathcal{D}_{\text{pool}}, i_{1:m-1}} \left[ \text{Var}\left[ w_m \mathcal{L}_{i_m} | i_{1:m-1}, \mathcal{D}_{\text{pool}} \right] \right]. \quad (6)$$

Although not obvious from inspection of (6), in Appendix B.6 we prove that the variance of $\tilde{R}_{\text{LURE}}$ is always less than that of $\tilde{R}_{\text{PURE}}$ subject to a mild assumption about the proposal which we detail there.

**Theorem 4.** *If Equation* (14) *in Appendix B.6 holds then* $\text{Var}[\tilde{R}_{LURE}] \leq \text{Var}[\tilde{R}_{PURE}]$. *If $M > 1$ and* $\mathbb{E}_{\mathcal{D}_{\text{pool}}} \left[ \text{Var}[w_m \mathcal{L}_{i_1} | \mathcal{D}_{\text{pool}}] \right] > 0$ *also hold, then the inequality is strict:* $\text{Var}[\tilde{R}_{LURE}] < \text{Var}[\tilde{R}_{PURE}]$.

To provide intuition into why this result holds, remember that $c_m$ were introduced to ensure that $\mathbb{E}[v_m]$ are all identically one. Therefore this weighting removes the tendency of $\tilde{R}_{\text{PURE}}$ to overemphasize the earlier samples; essentially increasing the effective sample size by correcting the imbalance.

We finish by confirming that $\tilde{R}_{\text{LURE}}$ is a consistent estimator as $M \rightarrow \infty$ (proof in Appendix B.7):

**Theorem 5.** *Under the same assumptions as Theorem 2:* $\lim_{M \rightarrow \infty} \mathbb{E}\left[ \left( \tilde{R}_{LURE} - r \right)^2 \right] = 0$.

## 3.3 From Active Learning Schemes to Proposals

We have introduced two elements of the active learning scheme: the risk estimators—$\tilde{R}_{\text{PURE}}$ and $\tilde{R}_{\text{LURE}}$—and the acquisition proposal distribution—$q(i_m | i_{1:m-1}, \mathcal{D}_{\text{pool}})$—which has so far remained general. So long as the acquisition proposal puts non-zero mass on all the training data, $\tilde{R}_{\text{PURE}}$ and

$\tilde{R}_{\text{LURE}}$ are unbiased and consistent as proven above. This is in contrast to the naive risk estimator $\tilde{R}$, for which the choice of proposal distribution affects the bias of the estimator.

It is easy to satisfy the requirement for non-zero mass everywhere. Even prior work which selects points deterministically (e.g., Bayesian Active Learning by Disagreement (BALD) (Houlsby et al., 2011) or a geometric heuristic like coreset construction (Sener & Savarese, 2018)) can be easily adapted. Any scheme, like BALD, that selects the points with `argmax` can use `softmax` to return a distribution. Alternatively, a distribution can be constructed analogous to epsilon-greedy exploration. With probability $\epsilon$ we pick uniformly, otherwise we pick the point returned by an arbitrary acquisition strategy. This adapts any deterministic active learning scheme to allow unbiased risk estimation.

It is also possible to use $\tilde{R}_{\text{LURE}}$ and $\tilde{R}_{\text{PURE}}$ with data collected using a proposal distribution that does not fully support the training data, though they will not fully correct the bias in this case. Namely, if we have a set of points, $I$, that are ignored by the proposal (i.e. that are assigned zero mass), we can still use $\tilde{R}_{\text{LURE}}$ and $\tilde{R}_{\text{PURE}}$ in the same way but they both introduce the same following bias:

$$\mathbb{E}[\tilde{R}^I_{\text{LURE}}] = \mathbb{E}[\tilde{R}^I_{\text{PURE}}] = \mathbb{E}\left[\mathbb{E}\left[\tilde{R}_{\text{LURE}}\middle|\mathcal{D}_{\text{pool}}\right] - \mathbb{E}\left[\frac{1}{N}\sum_{n\in I}\mathcal{L}_n\middle|\mathcal{D}_{\text{pool}}\right]\right] = r - \mathbb{E}\left[\frac{1}{N}\sum_{n\in I}\mathcal{L}_n\right].$$

Sometimes this bias will be small and may be acceptable if it enables a desired acquisition scheme, but in general one of the stochastic adaptations described above is likely to be preferable. One can naturally also extend this result to cases where $I$ varies at each iteration of the active learning (including deterministic acquisition strategies), for which we again have a non–zero bias.

Though the choices of acquisition proposal and risk estimator are algorithmically detached, choosing a good proposal will still be critical to performance in practice. In the next section, we will discuss how the proposal distribution can affect the *variance* of the estimators, and we will see that our approaches also offer the potential to reduce the variance of the naive biased estimator. Later, in §7, we will turn to a third element of active learning schemes—*generalization*—and consider the fact that optimization introduces a bias separately from the choice of risk estimator and proposal distribution.

## 4 UNDERSTANDING THE EFFECT OF $\tilde{R}_{\text{LURE}}$ AND $\tilde{R}_{\text{PURE}}$ ON VARIANCE

In order to show that the variance of our unbiased estimators can be *lower* than that of the biased $\tilde{R}$, with a well-chosen acquisition function, we first introduce an analogous result to Theorems 1 and 3 for $\tilde{R}$, the proof for which is given in Appendix B.8:

**Theorem 6.** *Let* $\mu_m := \mathbb{E}\left[\mathcal{L}_{i_m}\right]$ *and* $\mu_{m|i,\mathcal{D}} := \mathbb{E}\left[\mathcal{L}_{i_m}|i_{1:m-1}, \mathcal{D}_{\text{pool}}\right]$. *For* $\tilde{R}$ *(defined in* (1)*):*

$$\mathbb{E}\left[\tilde{R}\right] = \frac{1}{M}\sum_{m=1}^{M}\mu_m \quad (\neq r \text{ in general})$$

$$\text{Var}[\tilde{R}] = \overbrace{\text{Var}_{\mathcal{D}_{\text{pool}}}\left[\mathbb{E}\left[\tilde{R}\middle|\mathcal{D}_{\text{pool}}\right]\right]}^{\text{\textcircled{1}}} + \overbrace{\frac{1}{M^2}\sum_{m=1}^{M}\mathbb{E}_{\mathcal{D}_{\text{pool}},i_{1:m-1}}\left[\text{Var}\left[\mathcal{L}_{i_m}|i_{1:m-1},\mathcal{D}_{\text{pool}}\right]\right]}^{\text{\textcircled{2}}}$$
$$+ \frac{1}{M^2}\sum_{m=1}^{M}\underbrace{\mathbb{E}_{\mathcal{D}_{\text{pool}}}\left[\text{Var}\left[\mu_{m|i,\mathcal{D}}\middle|\mathcal{D}_{\text{pool}}\right]\right]}_{\text{\textcircled{3}}} + 2\underbrace{\mathbb{E}_{\mathcal{D}_{\text{pool}}}\left[\text{Cov}\left[\mathcal{L}_{i_m},\sum_{k<m}\mathcal{L}_{i_k}\middle|\mathcal{D}_{\text{pool}}\right]\right]}_{\text{\textcircled{4}}}. \tag{7}$$

Examining this expression suggests that the variances of $\tilde{R}_{\text{PURE}}$ and, in particular, $\tilde{R}_{\text{LURE}}$ will often be lower than that of $\tilde{R}$, given a suitable proposal. Consider the terms of (7): ① is analogous to the shared first term of (4) and (6), $\text{Var}\left[\mathcal{L}(\mathbf{y}, f_\theta(\mathbf{x}))\right]/N$. If $\tilde{R}$ were an unbiased estimator of $\hat{R}$ then these would be exactly equal, but the conditional bias introduced by $\tilde{R}$ also varies between pool datasets. In general, ① will typically be larger than, or similar to, its unbiased counterparts. In any case, recall that the first terms of (4) and (6) tend to be small contributors to the overall variance anyway, thus ① provides negligible scope for $\tilde{R}$ to provide notable benefits over our estimators.

We can also relate ② to terms in (4) and (6): it corresponds to the second half of (4), but where we replace of the expected conditional variances of the *weighted* losses $w_m\mathcal{L}_{i_m}$ with the unweighted losses $\mathcal{L}_{i_m}$. For effective proposals, $w_m$ and $\mathcal{L}_{i_m}$ should be anticorrelated: high loss points should have higher density and thus lower weight. This means the expected conditional variance of $w_m\mathcal{L}_{i_m}$ should be less than $\mathcal{L}_{i_m}$ for a well-designed acquisition strategy. Variation in the expected value

of the weights with $m$ can complicate this slightly for $\tilde{R}_{\text{PURE}}$, but the correction factors applied for $\tilde{R}_{\text{LURE}}$ avoids this issue and ensure that the second half of (6) will be reliably smaller than ②.

We have shown that the variance of $\tilde{R}_{\text{LURE}}$ is typically smaller than ① + ② under sensible proposals. Expression (7) has additional terms: ③ is trivially always positive and so contributes to higher variance for $\tilde{R}$ (it comes from variation in the bias in index sampling given $\mathcal{D}_{\text{pool}}$); ④ reflects correlations between the losses at different iterations which have been eliminated by our estimators. This term is harder to quantify and can be positive or negative depending on the problem. For example, sampling points without replacement can cause negative correlation, while the proposal adaptation itself can cause positive correlations (finding one high loss point can help find others). The former effect diminishes as $N$ grows, for fixed $M$, hinting that ④ may tend to be positive for $N \gg M$. Regardless, if ④ is big enough to change which estimator has higher variance then correlation between losses in different acquired points would lead to high bias in $\tilde{R}$.

In contrast, we prove in Appendix B.9 that under an optimal proposal distribution both $\tilde{R}_{\text{PURE}}$ and $\tilde{R}_{\text{LURE}}$ become exact estimators of the empirical risk for any number of samples $M$—such that they will inevitably have lower variance than $\tilde{R}$ in this case. A similar result holds when we are estimating gradients of the loss, though note that the optimal proposal is different in the two cases.

**Theorem 7.** *Given a non-negative loss, the optimal proposal distribution*

$$q^*(i_m; i_{1:m-1}, \mathcal{D}_{\text{pool}}) = \mathcal{L}_{i_m} / \Sigma_{n \notin i_{1:m-1}} \mathcal{L}_n$$

*yields estimators exactly equal to the pool risk, that is $\tilde{R}_{PURE} = \tilde{R}_{LURE} = \hat{R}$ almost surely $\forall M$.*

In practice, it is impossible to sample using the optimal proposal distribution. However, we make this point in order to prove that adopting our unbiased estimator is certainly capable of reducing variance relative to standard practice if appropriate acquisition strategies are used. It also provides interesting insights into what makes a good acquisition strategy from the perspective of the risk estimation itself.

## 5    RELATED WORK

Pool-based active learning (Lewis & Gale, 1994) is useful in cases where input data are prevalent but labeling them is expensive (Atlas et al., 1990; Settles, 2010). The bias from selective sampling was noted by MacKay (1992), but dismissed from a Bayesian perspective based on the likelihood principle. Others have noted that the likelihood principle remains controversial (Rainforth, 2017), and in this case would assume a well-specified model. Moreover, from a discriminative learning perspective this bias is uncontentiously problematic. Lowell et al. (2019) observe that active learning algorithms and datasets become coupled by active sampling and that datasets often outlive algorithms.

Despite the potential pitfalls, in deep learning this bias is generally ignored. As an informal survey, we examined the 15 most-cited peer-reviewed papers citing Gal et al. (2017b), which considered active learning to image data using neural networks. Of these, only two mentioned estimator bias but did not address it while the rest either ignored or were unaware of this problem (see Appendix D).

There have been some attempts to address active learning bias, but these have generally required fundamental changes to the active learning approach and only apply to particular setups. Beygelzimer et al. (2009), Chu et al. (2011), and (Cortes et al., 2019) apply importance-sampling corrections (Sugiyama, 2006; Bach, 2006) to *online* active learning. Unlike pool-based active learning, this involves deciding whether or not to sample a new point as it arrives from an infinite distribution. This makes importance-sampling estimators much easier to develop, but as Settles (2010) notes, "*the pool-based scenario appears to be much more common among application papers.*"

Ganti & Gray (2012) address unbiased active learning in a pool-based setting by sampling from the pool *with replacement*. This effectively converts pool-based learning into a stationary online learning setting, although it overweights data that happens to be sampled early. Sampling with replacement is unwanted in active learning because it requires retraining the model on duplicate data which is either impossible or wasteful depending on details of the setting. Moreover, they only prove the consistency of their estimator under very strong assumptions (well-specified linear models with noiseless labels and a mean-squared-error loss). Imberg et al. (2020) consider optimal proposal distributions in an importance-sampling setting. Outside the context of active learning, Byrd & Lipton (2019) question the value of importance-weighting for deep learning, which aligns with our findings below.

# 6  APPLYING $\tilde{R}_{\text{LURE}}$ AND $\tilde{R}_{\text{PURE}}$

We first verify that $\tilde{R}_{\text{LURE}}$ and $\tilde{R}_{\text{PURE}}$ remove the bias introduced by active learning and examine the variance of the estimators. We do this by taking a fixed function whose parameters are independent of $\mathcal{D}_{\text{train}}$ and estimating the risk using actively sampled points. We note that this is equivalent to the problem of estimating the risk of an already trained model in a sample-efficient way given unlabelled test data. We consider two settings: an inflexible model (linear regression) on toy but non-linear data and an overparameterized model (convolutional Bayesian neural network) on a modified version of MNIST with unbalanced classes and noisy labels.

**Linear regression.** For linear functions, removing active learning bias (ALB), i.e., the statistical bias introduced by active learning, is critical. We illustrate this in Figure 1. Actively sampled points overrepresent unusual parts of the distribution, so a model learned using the unweighed $\mathcal{D}_{\text{train}}$ differs

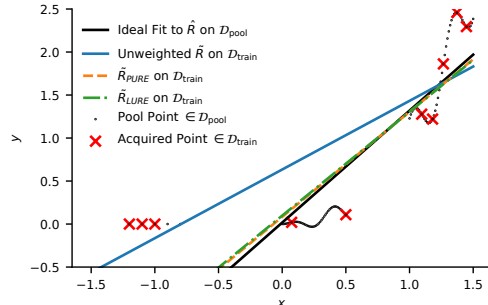

Figure 1: **Illustrative linear regression.** Active learning deliberately over-samples unusual points (red x's) which no longer match the population (black dots). Common practice uses the biased unweighted estimator $\tilde{R}$ which puts too much emphasis on unusual points. Our unbiased estimators $\tilde{R}_{\text{PURE}}$ and $\tilde{R}_{\text{LURE}}$ fix this, learning a function using only $\mathcal{D}_{\text{train}}$ nearly equal to the ideal you would get if you had labels for the whole of $\mathcal{D}_{\text{pool}}$, despite only using a few points.

from the ideal function fit to the whole of $\mathcal{D}_{\text{pool}}$. Using our corrective weights more closely approximates the ideal line. The full details of the population distribution and geometric acquisition proposal distribution are in Appendix C.1, where we also show results using an alternative epsilon-greedy proposal. We inspect the ALB in Figure 2a by comparing the estimated risk (with squared error loss and a fixed function) to the corresponding true population risk $\hat{R}$. While $M < N$, the unweighted $\tilde{R}$ is biased (in practice we never have $M = N$ as then actively learning is unnecessary). $\tilde{R}_{\text{PURE}}$ and $\tilde{R}_{\text{LURE}}$ are unbiased throughout. However, they have high variance because the proposal is rather poor. Shading represents the std. dev. of the bias over 1000 different acquisition trajectories.

**Bayesian Neural Network.** We actively classify MNIST and FashionMNIST images using a convolutional Bayesian neural network (BNN) with roughly 80,000 parameters. In Figure 2b and 2c we show that $\tilde{R}_{\text{PURE}}$ and $\tilde{R}_{\text{LURE}}$ remove the ALB. Here the variance of $\tilde{R}_{\text{PURE}}$ and $\tilde{R}_{\text{LURE}}$ is *lower* or similar to the biased estimator. This is because the acquisition proposal distribution, a stochastic relaxation of the Bayesian Active Learning by Disagreement (BALD) objective (Houlsby et al., 2011), is effective (c.f. §4). A full description of the dataset and procedure is provided in Appendix C.2. Our modified MNIST dataset is unbalanced and has noisy labels, which makes the bias more distorting.

Overall, Figure 2 shows that our estimators remove the bias introduced by active learning, as expected, and can do so with reduced variance given an acquisition proposal distribution that puts a high probability mass on more informative/surprising high-expected-loss points.

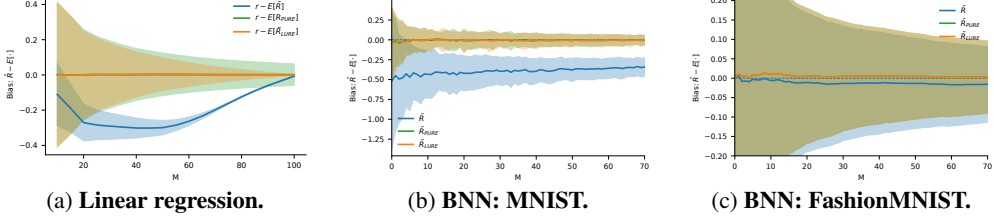

(a) **Linear regression.**   (b) **BNN: MNIST.**   (c) **BNN: FashionMNIST.**

Figure 2: $\tilde{R}_{\text{PURE}}$ and $\tilde{R}_{\text{LURE}}$ remove bias introduced by active learning, while unweighted $\tilde{R}$, which most active learning work uses, is biased. Note the sign: $\tilde{R}$ *overestimates* risk because active learning samples the hardest points. Variance for $\tilde{R}_{\text{PURE}}$ and $\tilde{R}_{\text{LURE}}$ depends on the acquisition distribution placing high weight on high-expected-loss points. In (b), the BALD-style distribution means that the variance of the unbiased estimators is *smaller*. For FashionMNIST, (c), active learning bias is small and high variance in all cases. Shading is $\pm 1$ standard deviation.

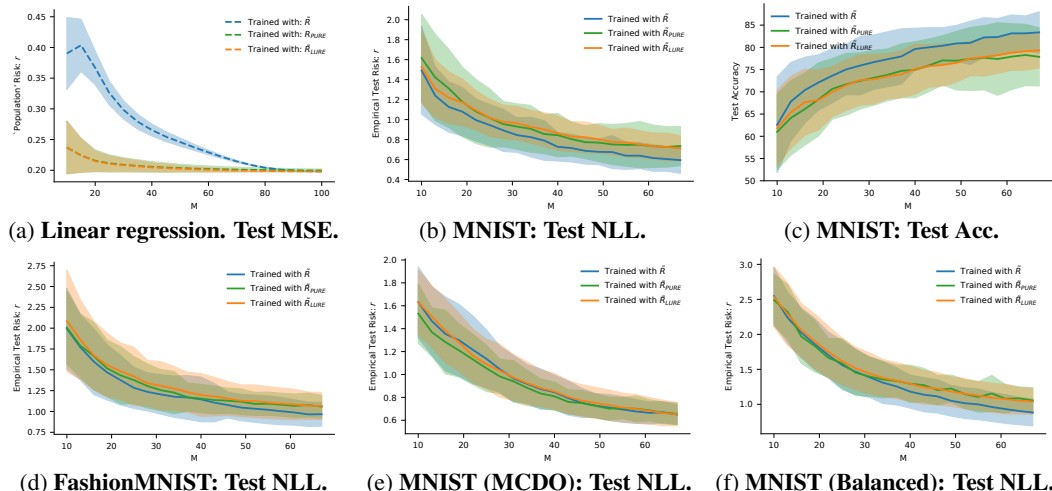

(a) **Linear regression. Test MSE.**    (b) **MNIST: Test NLL.**    (c) **MNIST: Test Acc.**

(d) **FashionMNIST: Test NLL.**    (e) **MNIST (MCDO): Test NLL.**    (f) **MNIST (Balanced): Test NLL.**

Figure 3: For linear regression, the models trained with $\tilde{R}_{\text{PURE}}$ or $\tilde{R}_{\text{LURE}}$ have lower 'population' risk. In contrast, BNNs trained with $\tilde{R}_{\text{LURE}}$ or $\tilde{R}_{\text{PURE}}$ perform either similarly (e) or slightly worse (b,c,d,f), even though they remove bias and have lower variance. Shading is one standard deviation. For (a) 1000 samples and '$r$' estimated on 10,100 points from distribution. For (b)/(c) 45 samples and '$r$' estimated on the test dataset. (d)-(f) are alternative settings to validate consistency of result.

Next, we examine the overall effect of using the unbiased estimators to *learn* a model on *downstream performance*. Intuitively, removing bias in training while also reducing the variance ought to improve the downstream task objective: test loss and accuracy. To investigate this, we train models using $\tilde{R}$, $\tilde{R}_{\text{LURE}}$, and $\tilde{R}_{\text{PURE}}$ with actively sampled data and measure the population risk of each model.

For linear regression (Figure 3a), the new estimators improve the test loss—even with small numbers of acquired points we have nearly optimal test loss (estimated with many samples). However, for the BNN, there is a small but significant negative impact on the full test dataset loss of training with $\tilde{R}_{\text{LURE}}$ or $\tilde{R}_{\text{PURE}}$ (Figure 3b) and a slightly larger negative impact on test accuracy (Figure 3c). That is, we get a better model by training using a biased estimator with higher variance!

To validate this further, we consider models trained instead on FashionMNIST (Fig. 3d), on MNIST but with Monte Carlo dropout (MCDO) (Gal & Ghahramani, 2015) (Fig. 3e), and on a balanced version of the MNIST data (Fig. 3f). In all cases we find similar patterns, suggesting the effects are not overly sensitive to the setting. Further experiments and ablations can be found in Appendix C.2.

## 7 ACTIVE LEARNING BIAS IN THE CONTEXT OF OVERALL BIAS

In order to explain the finding that $\tilde{R}_{\text{LURE}}$ hurts training for the BNN, we return to the bias introduced by overfitting, allowing us to examine the effect of removing statistical bias in the context of *overall* bias. Namely, we need to consider the fact that training would induce am overfitting bias (OFB) even if we had not used active learning. If we optimize parameters $\theta$ according to $\hat{R}$, then $\mathbb{E}[\hat{R}(\theta^*)] \neq r$, because the optimized parameters $\theta^*$ tend to explain training data better than unseen data. Using $\tilde{R}_{\text{LURE}}$, which removes *statistical bias*, we can isolate OFB in an active learning setting. More formally, supposing we are optimizing any of the discussed risk estimators (which we will write using $\tilde{R}_{(\cdot)}$ as a placeholder to stand for any of them) we define the OFB as:

$$B_{\text{OFB}}(\tilde{R}_{(\cdot)}) = r - \tilde{R}_{\text{LURE}}(\theta^*) \quad \text{where} \quad \theta^* = \arg\min_\theta(\tilde{R}_{(\cdot)})$$

$B_{\text{OFB}}(\tilde{R}_{(\cdot)})$ depends on the details of the optimization algorithm and the dataset. Understanding it fully means understanding generalization in machine learning and is outside our scope. We can still gain insight into the interaction of active learning bias (ALB) and OFB. Consider the possible relationships between the magnitudes of ALB and OFB: [**ALB >> OFB**] Removing ALB reduces overall bias and is most likely to occur when $f_\theta$ is not very expressive such that there is little chance of overfitting. [**ALB << OFB**] Removing ALB is irrelevant as model has massively overfit regardless. [**ALB ≈ OFB**] Here *sign* is critical. If ALB and OFB have opposite signs and similar scale, they

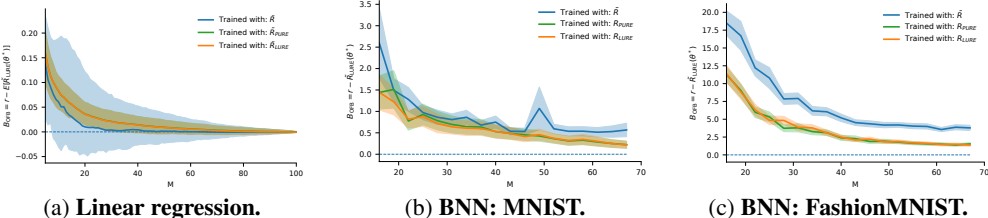

(a) **Linear regression.**   (b) **BNN: MNIST.**   (c) **BNN: FashionMNIST.**

Figure 4: Overfitting bias—$B_{\text{OFB}}$—for models trained using the three objectives. (a) Linear regression, $B_{\text{OFB}}$ is small compared to ALB (c.f. Figure 2a). Shading IQR. 1000 trajectories. (b) BNN, $B_{\text{OFB}}$ is similar scale and opposite magnitude to ALB (c.f. Figure 2b). (c) BNN on FashionMNIST, OFB is somewhat larger than with MNIST, particularly for $\tilde{R}$ (i.e. our approaches reduce overfitting) and dominates active learning bias (c.f. Figure 2c). Shading $\pm 1$ standard error. 150 trajectories.

will tend to cancel each other out. Indeed, they usually have opposite signs. $B_{\text{OFB}}$ is usually positive: $\theta^*$ fits the training data better than unseen data. ALB is generally negative: we actively choose unusual/surprising/informative points which are harder to fit than typical points.

Therefore, when significant overfitting is possible, unless ALB is also large, removing ALB will have little effect and can *even be harmful*. This hypothesis would explain the observations in §6 if we were to show that $B_{\text{OFB}}$ was small for linear regression but had a similar magnitude and opposite sign to ALB for the BNN. This is exactly what we show in Figure 4.

Specifically, we see that for linear regression, the $B_{\text{OFB}}$ for models trained with $\tilde{R}$, $\tilde{R}_{\text{PURE}}$, and $\tilde{R}_{\text{LURE}}$ are all small (Figure 4a) when contrasted to the ALB shown in Figure 2a. Here ALB $>>$ OFB; removing ALB matters. For BNNs we instead see that the OFB has opposite sign to the ALB but is either similar in scale for MNIST (Figures 2b and 4b), or the OFB is much larger than ALB for Fashion MNIST (Figures 4c and 2c). The two sources of bias thus (partially) cancel out. Essentially, using active learning can be treated (quite instrumentally) as an ad hoc form of regularization. This explains why removing ALB can hurt active learning with neural networks.

## 8 CONCLUSIONS

Active learning is a powerful tool but raises potential problems with statistical bias. We offer a corrective weighting which removes that bias with negligible compute/memory costs and few requirements—it suits standard pool–based active learning without replacement. It requires a non–zero proposal distribution over all unlabelled points but existing acquisition functions can be easily transformed into sampling distributions. Indeed, estimates of scores like mutual information are so noisy that many applications already have an implicit proposal distribution.

We show that removing active learning bias (ALB) can be helpful in some settings, like linear regression, where the model is not sufficiently complex to perfectly match the data, such that the exact loss function and input data distribution are essential in discriminating between different possible (imperfect) model fits. We also find that removing ALB can be counter-productive for overparameterized models like neural networks, even if its removal also reduces the variance of the estimators, because here the ALB can help cancel out the bias originating from overfitting. This leads to the interesting conclusion that active learning can be helpful not only as a mechanism to reduce variance as it was originally designed, but also because it introduces a bias that can be *actively helpful* by regularizing the model. This helps explain why active learning with neural networks has shown success despite using a biased risk estimator.

We propose the following rules of thumb for deciding when to embrace or correct the bias, noting that we should always prefer $\tilde{R}_{\text{LURE}}$ to $\tilde{R}_{\text{PURE}}$. First, the more closely the acquisition proposal distribution approaches the optimal distribution (as per Theorem 7), the relatively better $\tilde{R}_{\text{LURE}}$ will be to $\tilde{R}$. Second, the less overfitting we expect, the more likely it is that $\tilde{R}_{\text{LURE}}$ will be useful as it reduces the chance that the ALB will actually help. Third, $\tilde{R}_{\text{LURE}}$ will tend to have more of an effect for highly imbalanced datasets, as the biased estimator will over-represent actively selected but unlikely datapoints. Fourth, if the training data does not accurately represent the test data, using $\tilde{R}_{\text{LURE}}$ will likely be less important as the ALB will tend to be dwarfed by bias from the distribution shift. Fifth, at test–time, where optimization and overfitting bias are no-longer an issue, there is little cost to using $\tilde{R}_{\text{LURE}}$ to evaluate a model and it will usually be beneficial. This final application, of active learning for model evaluation, is an interesting new research direction that is opened up by our estimators.

## ACKNOWLEDGEMENTS

The authors would like to especially thank Lewis Smith for his helpful conversations and specifically for his assistance with the proof of Theorem 4. In addition, we would like to thank for their conversations and advice Joost van Amersfoort and Andreas Kirsch.

The authors are grateful to the Engineering and Physical Sciences Research Council for their support of the Centre for Doctoral Training in Cyber Security, University of Oxford as well as the Alan Turing Institute.

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

## A  OVERVIEW OF ACTIVE LEARNING

Active learning selectively picks datapoints for which to acquire labels with the aim of more sample-efficient learning. For an excellent overview of the general active learning problem setting we refer the reader to Settles (2010).

Since that review was written, a number of significant advances have further developed active learning. Houlsby et al. (2011) develop an efficient way to estimate the mutual information between model parameters and the output distribution, which can be used for the Bayesian Active Learning by Disagreement (BALD) score.

Active learning has been applied to deep learning, especially for vision Gal et al. (2017b); Wang et al. (2017). In neural networks specifically, empirical work has suggested that simple geometric core-set style approaches can outperform uncertainty-based acquisition functions (Sener & Savarese, 2018).

A lot of recent work in active learning has focused on speeding up acquisition from a computational perspective (Coleman et al., 2020) and allowing batch acquisition in order to parallelize labelling (Kirsch et al., 2019; Ash et al., 2020). Some work has also focused on applying active learning to specific settings with particular constraints (Krishnamurthy et al., 2017; Yan et al., 2018; Sundin et al., 2019; Behpour et al., 2019; Shi & Yu, 2019; Hu et al., 2019).

## B  PROOFS

### B.1  PROOF OF LEMMA 1

**Lemma 1.** *The individual terms $a_m$ of $\tilde{R}_{PURE}$ are unbiased estimators of the risk:* $\mathbb{E}\left[a_m\right] = r$.

*Proof.* We begin by applying the tower property of expectations:

$$\mathbb{E}\left[a_m\right] = \mathbb{E}\left[w_m\mathcal{L}_{i_m} + \frac{1}{N}\sum_{t=1}^{m-1}\mathcal{L}_{i_t}\right]$$

$$= \mathbb{E}_{\mathcal{D}_{\text{pool}},i_{1:m-1}}\left[\mathbb{E}_{i_m}\left[w_m\mathcal{L}_{i_m} + \frac{1}{N}\sum_{t=1}^{m-1}\mathcal{L}_{i_t}\,\middle|\,\mathcal{D}_{\text{pool}},i_{1:m-1}\right]\right].$$

By further noting that $\mathbb{E}_{i_m}\left[w_m\mathcal{L}_{i_m}\,|\,\mathcal{D}_{\text{pool}},i_{1:m-1}\right]$ can be written out analytically as a sum over all the possible values of $i_m$, while $\frac{1}{N}\sum_{t=1}^{m-1}\mathcal{L}_{i_t}$ is deterministic given $\mathcal{D}_{\text{pool}}$ and $i_{1:m-1}$ we have:

$$= \mathbb{E}_{\mathcal{D}_{\text{pool}},i_{1:m-1}}\left[\sum_{n\notin i_{1:m-1}}\left(\cancel{q(i_m;i_{1:m-1},\mathcal{D}_{\text{pool}})}\frac{\mathcal{L}_{i_m}}{N\,\cancel{q(i_m;i_{1:m-1},\mathcal{D}_{\text{pool}})}}\right) + \frac{1}{N}\sum_{t=1}^{m-1}\mathcal{L}_{i_t}\right]$$

$$= \mathbb{E}_{\mathcal{D}_{\text{pool}},i_{1:m-1}}\left[\frac{1}{N}\sum_{n=1}^{N}\mathcal{L}_n\right],$$

But $\mathcal{L}_n$ is now independent of the indices which have been sampled:

$$= \mathbb{E}_{\mathcal{D}_{\text{pool}}}\left[\frac{1}{N}\sum_{n=1}^{N}\mathcal{L}_n\right] = \mathbb{E}_{\mathcal{D}_{\text{pool}}}\left[\hat{R}\right] = r.$$

□

### B.2  PROOF OF THE UNBIASEDNESS AND VARIANCE FOR $\tilde{R}_{\text{PURE}}$: THEOREM 1

**Theorem 1.** $\tilde{R}_{PURE}$ *as defined above has the properties:*

$$\mathbb{E}\left[\tilde{R}_{PURE}\right] = r,$$

$$\text{Var}\left[\tilde{R}_{PURE}\right] = \frac{\text{Var}\left[\mathcal{L}(\mathbf{y}, f_\theta(\mathbf{x}))\right]}{N} + \frac{1}{M^2}\sum_{m=1}^{M}\mathbb{E}_{\mathcal{D}_{\text{pool}},i_{1:m-1}}\left[\text{Var}\left[w_m\mathcal{L}_{i_m}|i_{1:m-1},\mathcal{D}_{\text{pool}}\right]\right]. \quad (4)$$

*Proof.* Having established Lemma 1, unbiasedness follows quickly from the linearity of expectations:

$$\mathbb{E}\left[\tilde{R}_{\text{PURE}}\right] = \mathbb{E}\left[\frac{1}{M}\sum_{m=1}^{M} a_m\right] = \frac{1}{M}\sum_{m=1}^{M}\mathbb{E}\left[a_m\right] = r.$$

For the variance we instead have (starting with the definition of $\tilde{R}_{\text{PURE}}$):

$$\text{Var}\left[\tilde{R}_{\text{PURE}}\right] = \mathbb{E}\left[\left(\frac{1}{M}\sum_{m=1}^{M} a_m\right)^2\right] - r^2,$$

which by the tower property of expectations

$$= \mathbb{E}\left[\mathbb{E}\left[\left(\frac{1}{M}\sum_{m=1}^{M} a_m\right)^2\middle|\mathcal{D}_{\text{pool}}\right]\right] - r^2,$$

and from the definition of variance

$$= \mathbb{E}\left[\text{Var}\left[\frac{1}{M}\sum_{m=1}^{M} a_m\middle|\mathcal{D}_{\text{pool}}\right] + \hat{R}^2\right] - r^2,$$

$$= \mathbb{E}\left[\text{Var}\left[\frac{1}{M}\sum_{m=1}^{M} a_m\middle|\mathcal{D}_{\text{pool}}\right]\right] + \text{Var}\left[\hat{R}\right],$$

which using the fact that $\hat{R}$ is a standard Monte Carlo estimator

$$= \mathbb{E}\left[\text{Var}\left[\frac{1}{M}\sum_{m=1}^{M} a_m\middle|\mathcal{D}_{\text{pool}}\right]\right] + \frac{\text{Var}\left[\mathcal{L}(\mathbf{y}, f_\theta(\mathbf{x}))\right]}{N}, \tag{8}$$

where $\mathbf{x}, \mathbf{y} \sim p_{\text{data}}$. Now considering the first term we have

$$\text{Var}\left[\frac{1}{M}\sum_{m=1}^{M} a_m\middle|\mathcal{D}_{\text{pool}}\right] = \mathbb{E}\left[\left(\frac{1}{M}\sum_{m=1}^{M} a_m - \hat{R}\right)^2\middle|\mathcal{D}_{\text{pool}}\right]$$

$$= \frac{1}{M^2}\sum_{m=1}^{M}\sum_{k=1}^{M}\mathbb{E}\left[\left(a_m - \hat{R}\right)\left(a_k - \hat{R}\right)\middle|\mathcal{D}_{\text{pool}}\right]. \tag{9}$$

We attack this term by first considering the terms for which $m \neq k$ and show that these yield $\mathbb{E}\left[(a_m - r)(a_k - r)|\mathcal{D}_{\text{pool}}\right] = 0$, returning to the $m = k$ terms later. We will assume, without loss of generality, that $k < m$, noting that by symmetry the same set of arguments can be similarly applied when $m < k$. Substituting in the definition of $a_m$ from equation (2):

$$\mathbb{E}\left[(a_m - \hat{R})(a_k - \hat{R})\middle|\mathcal{D}_{\text{pool}}\right]$$

$$= \mathbb{E}\left[\left(w_m\mathcal{L}_{i_m} + \frac{1}{N}\sum_{t=1}^{m-1}\mathcal{L}_{i_t} - \hat{R}\right)\left(w_k\mathcal{L}_{i_k} + \frac{1}{N}\sum_{s=1}^{k-1}\mathcal{L}_{i_s} - \hat{R}\right)\middle|\mathcal{D}_{\text{pool}}\right].$$

We introduce the notation $\hat{R}_m^{rem} = \hat{R} - \frac{1}{N}\sum_{t=1}^{m-1}\mathcal{L}_{i_t}$ to describe the remainder of the empirical risk ascribable to the datapoint with index $i_m$. Then, by multiplying out the terms we have:

$$\mathbb{E}\left[(a_m - \hat{R})(a_k - \hat{R})\middle|\mathcal{D}_{\text{pool}}\right] = \underbrace{\mathbb{E}\left[w_m\mathcal{L}_{i_m} w_k\mathcal{L}_{i_k}|\mathcal{D}_{\text{pool}}\right]}_{\text{\textcircled{a}}} - \underbrace{\mathbb{E}\left[w_m\mathcal{L}_{i_m}\hat{R}_k^{rem}\middle|\mathcal{D}_{\text{pool}}\right]}_{\text{\textcircled{b}}}$$

$$- \underbrace{\mathbb{E}\left[w_k\mathcal{L}_{i_k}\hat{R}_m^{rem}\middle|\mathcal{D}_{\text{pool}}\right]}_{\text{\textcircled{c}}} + \underbrace{\mathbb{E}\left[\hat{R}_k^{rem}\hat{R}_m^{rem}\middle|\mathcal{D}_{\text{pool}}\right]}_{\text{\textcircled{d}}}.$$

Now by the tower property:

$$\text{\textcircled{a}} = \mathbb{E}\left[\mathbb{E}\left[w_m\mathcal{L}_{i_m} w_k\mathcal{L}_{i_k}\,|\,\mathcal{D}_{\text{pool}}, i_{1:m-1}\right]|\mathcal{D}_{\text{pool}}\right],$$

and noting that because $k < m$, $w_k \mathcal{L}_{i_k}$ is deterministic given $\mathcal{D}_{\text{pool}}$ and $i_{1:m-1}$:

$$= \mathbb{E}\left[ w_k \mathcal{L}_{i_k} \mathbb{E}\left[ w_m \mathcal{L}_{i_m} \,\middle|\, \mathcal{D}_{\text{pool}}, i_{1:m-1} \right] \middle| \mathcal{D}_{\text{pool}} \right]$$

$$= \mathbb{E}\left[ w_k \mathcal{L}_{i_k} \hat{R}_m^{rem} \middle| \mathcal{D}_{\text{pool}} \right],$$

which thus cancels with $\text{\textcircled{c}}$.

The $\text{\textcircled{b}}$ and $\text{\textcircled{d}}$ cancel similarly:

$$\text{\textcircled{b}} = \mathbb{E}\left[ \mathbb{E}\left[ w_m \mathcal{L}_{i_m} \hat{R}_k^{rem} \,\middle|\, \mathcal{D}_{\text{pool}}, i_{1:m-1} \right] \middle| \mathcal{D}_{\text{pool}} \right]$$

$$= \mathbb{E}\left[ \hat{R}_k^{rem} \mathbb{E}\left[ w_m \mathcal{L}_{i_m} \,\middle|\, \mathcal{D}_{\text{pool}}, i_{1:m-1} \right] \middle| \mathcal{D}_{\text{pool}} \right]$$

$$= \mathbb{E}\left[ \hat{R}_k^{rem} \hat{R}_m^{rem} \middle| \mathcal{D}_{\text{pool}} \right] = \text{\textcircled{d}}.$$

Putting this together, we have that:

$$\mathbb{E}\left[ (a_m - \hat{R})(a_k - \hat{R}) \middle| \mathcal{D}_{\text{pool}} \right] = 0 \quad \forall k \neq m.$$

Considering now the $m = k$ terms we have

$$\mathbb{E}\left[ \left(a_m - \hat{R}\right)^2 \middle| \mathcal{D}_{\text{pool}} \right] = \mathbb{E}_{i_{1:m-1}}\left[ \mathbb{E}_{i_m}\left[ \left(a_m - \hat{R}\right)^2 \middle| i_{1:m-1}, \mathcal{D}_{\text{pool}} \right] \middle| \mathcal{D}_{\text{pool}} \right]$$

$$= \mathbb{E}_{i_{1:m-1}}\left[ \text{Var}\left[ a_m | i_{1:m-1}, \mathcal{D}_{\text{pool}} \right] | \mathcal{D}_{\text{pool}} \right]$$

$$= \mathbb{E}_{i_{1:m-1}}\left[ \text{Var}\left[ w_m \mathcal{L}_{i_m} | i_{1:m-1}, \mathcal{D}_{\text{pool}} \right] | \mathcal{D}_{\text{pool}} \right].$$

Finally substituting everything back into (9) and then (8), and applying the tower property gives

$$\text{Var}\left[ \tilde{R}_{\text{PURE}} \right] = \frac{\text{Var}\left[ \mathcal{L}(\mathbf{y}, f_\theta(\mathbf{x})) \right]}{N} + \frac{1}{M^2} \sum_{m=1}^{M} \mathbb{E}_{\mathcal{D}_{\text{pool}}, i_{1:m-1}}\left[ \text{Var}\left[ w_m \mathcal{L}_{i_m} | i_{1:m-1}, \mathcal{D}_{\text{pool}} \right] \right]$$

and we are done. $\qquad\square$

## B.3 Proof of the consistency of $\tilde{R}_{\text{PURE}}$: Theorem 2

**Theorem 2.** *Let $\alpha = N/M$ and assume that $\alpha > 1$. If $\mathbb{E}\left[ \mathcal{L}(\mathbf{y}, f_\theta(\mathbf{x}))^2 \right] < \infty$ and*

$$\exists \beta > 0 \; : \; \min_{n \in \{1:N \setminus i_{1:m-1}\}} q(i_m = n; i_{1:m-1}, \mathcal{D}_{\text{pool}}) \geq \beta/N \quad \forall N \in \mathbb{Z}^+, m \leq N,$$

*then $\tilde{R}_{PURE}$ converges in its $L^2$ norm to $r$ as $M \to \infty$, i.e., $\lim_{M \to \infty} \mathbb{E}\left[ (\tilde{R}_{PURE} - r)^2 \right] = 0$.*

*Proof.* Theorem 1 showed that $\tilde{R}_{\text{PURE}}$ is an unbiased estimator and so we first note that its MSE is simply its variance, which we found in (4). Substituting $N = \alpha M$:

$$\mathbb{E}\left[ \left( \tilde{R}_{\text{PURE}} - r \right)^2 \right] = \text{Var}\left[ \tilde{R}_{\text{PURE}} \right]$$

$$= \frac{\text{Var}\left[ \mathcal{L}(\mathbf{y}, f_\theta(\mathbf{x})) \right]}{\alpha M} + \frac{1}{M^2} \sum_{m=1}^{M} \mathbb{E}_{\mathcal{D}_{\text{pool}}, i_{1:m-1}}\left[ \text{Var}\left[ w_m \mathcal{L}_{i_m} | i_{1:m-1}, \mathcal{D}_{\text{pool}} \right] \right].$$

The first term tends to zero as $M \to \infty$ as our standard assumptions guarantee that $\frac{1}{\alpha} \text{Var}\left[ \mathcal{L}(\mathbf{y}, f_\theta(\mathbf{x})) \right] < \infty$. For the second term we note that our assumptions about $q$ guarantee that $w_m \leq 1/\beta$ and thus:

$$\text{Var}\left[ w_m \mathcal{L}_{i_m} | i_{1:m-1}, \mathcal{D}_{\text{pool}} \right] = \mathbb{E}\left[ w_m^2 \mathcal{L}_{i_m}^2 | i_{1:m-1}, \mathcal{D}_{\text{pool}} \right] - \left( \hat{R} - \frac{1}{M} \sum_{t=1}^{m-1} \mathcal{L}_{i_t} \right)^2$$

$$\leq \frac{1}{\beta^2} \mathbb{E}\left[ \mathcal{L}_{i_m}^2 | i_{1:m-1}, \mathcal{D}_{\text{pool}} \right] - \left( \hat{R} - \frac{1}{M} \sum_{t=1}^{m-1} \mathcal{L}_{i_t} \right)^2$$

$$< \infty \quad \forall i_{1:m-1}, \mathcal{D}_{\text{pool}}$$

as our assumptions guarantee that $\frac{1}{\beta^2} < \infty$, we have $\mathbb{E}_{i_m}\left[\mathcal{L}_{i_m}^2\right] < \infty$ and so the empirical risk and losses are finite.

Given that each $\text{Var}\left[w_m \mathcal{L}_{i_m} | i_{1:m-1}, \mathcal{D}_{\text{pool}}\right]$ is finite, it follows that:

$$s^2 := \frac{1}{M} \sum_{m=1}^{M} \mathbb{E}_{\mathcal{D}_{\text{pool}}, i_{1:m-1}}\left[\text{Var}\left[w_m \mathcal{L}_{i_m} | i_{1:m-1}, \mathcal{D}_{\text{pool}}\right]\right] < \infty,$$

and we thus have:

$$\lim_{M \to \infty} \mathbb{E}\left[\left(\tilde{R}_{\text{PURE}} - R\right)^2\right] = \lim_{M \to \infty} \left(\frac{\text{Var}\left[\mathcal{L}(\mathbf{y}, f_\theta(\mathbf{x}))\right]}{\alpha M} + \frac{s^2}{M}\right) = 0$$

as desired. $\qquad \square$

## B.4 Derivation of the Constants of $\tilde{R}_{\text{LURE}}$

We note from before that because of the unbiasedness of $a_m$:

$$\mathbb{E}\left[C \sum_{m=1}^{M} c_m a_m\right] = r \quad \text{where} \quad C = \frac{1}{\sum_{m=1}^{M} c_m},$$

To construct our improved estimator $\tilde{R}_{\text{LURE}}$, we now need to find the right constants $c_m$, that in turn lead to overall weights $v_m$ (as per (5)) such that $\mathbb{E}\left[v_m\right] = 1$. We start by substituting in the definition of $a_m$:

$$\tilde{R}_{\text{LURE}} := C \sum_{m=1}^{M} v_m \mathcal{L}_{i_m} := C \sum_{m=1}^{M} c_m a_m = C \sum_{m=1}^{M} \left(c_m w_m \mathcal{L}_{i_m} + \frac{c_m}{N} \sum_{t=1}^{m-1} \mathcal{L}_{i_t}\right),$$

and then redistributing the $\mathcal{L}_{i_t}$ from later terms where they match $m$:

$$v_m = c_m w_m + \frac{1}{N} \sum_{t=m+1}^{M} c_t.$$

Note that though $\tilde{R}_{\text{LURE}}$ remains an unbiased estimator of the risk, each individual term $v_m \mathcal{L}_{i_m}$ is not. Now we require: $\mathbb{E}\left[v_m\right] = 1 \quad \forall m \in \{1, ..., M\}$. Remembering that $w_m = 1/(Nq(i_m; i_{1:m-1}, \mathcal{D}_{\text{pool}}))$:

$$\mathbb{E}\left[v_m\right] = \frac{c_m}{N} \mathbb{E}\left[\frac{1}{q(i_m; i_{1:m-1}, \mathcal{D}_{\text{pool}})}\right] + \frac{1}{N} \sum_{t=m+1}^{M} c_t$$

$$= \frac{c_m}{N} \sum_{n \notin i_{1:m-1}} \frac{q(i_m = n; i_{1:m-1}, \mathcal{D}_{\text{pool}})}{q(i_m = n; i_{1:m-1}, \mathcal{D}_{\text{pool}})} + \frac{1}{N} \sum_{t=m+1}^{M} c_t$$

$$= \frac{c_m(N - m + 1)}{N} + \frac{1}{N} \sum_{t=m+1}^{M} c_t.$$

Imposing that each $\mathbb{E}\left[v_m\right] = 1$, we now have $M$ equations for $M$ unknowns $c_1, \ldots, c_M$, such that we can find the required values of $c_m$ by solving the set of simultaneous equations:

$$(N - m + 1)\frac{c_m}{N} + \frac{1}{N} \sum_{t=m+1}^{M} c_t = 1 \quad \forall m \in \{1, \ldots, M\}. \tag{10}$$

We do this by induction. First, consider $\mathbb{E}\left[v_m\right] - \mathbb{E}\left[v_{m+1}\right] = 0$, for which can be rewritten:

$$(N - m + 1)\frac{c_m}{N} - (N - m)\frac{c_{m+1}}{N} + \frac{c_{m+1}}{N} = 0$$

and thus:

$$c_m = \frac{N - m - 1}{N - m + 1}c_{m+1}. \tag{11}$$

By further noting that the solution for $m = M$ is trivial:

$$c_M = \frac{N}{N - M + 1},$$

we have by induction

$$c_m = \frac{N}{N - M + 1} \prod_{t=m}^{M-1} \frac{N - t - 1}{N - t + 1}$$

$$= \frac{N}{N - M + 1} \exp\left(\sum_{t=m}^{M-1} \log(N - t - 1) - \log(N - t + 1)\right).$$

Now we can exploit the fact that there is a canceling of most of the term in this sum. The exceptions are $-\log(N - m + 1)$, $-\log(N - m)$, $\log(N - M + 1)$, and $\log(N - M)$. We thus have:

$$c_m = \frac{N}{N - M + 1} \exp\left(\log(N - M) + \log(N - M + 1) - \log(N - m) - \log(N - m + 1)\right)$$

$$= \frac{N(N - M)}{(N - m)(N - m + 1)}.$$

which is our final simple form for $c_m$. We can now check that this satisfies the required recursive relationship (noting it trivially gives the correct value for $c_M$) as per (11):

$$c_m = \left(\frac{N - m - 1}{N - m + 1}\right) c_{m+1}$$

$$= \left(\frac{N - m - 1}{N - m + 1}\right) \frac{N(N - M)}{(N - m - 1)(N - m)}$$

$$= \frac{N(N - M)}{(N - m)(N - m + 1)}$$

as required. Similarly, we can straightforwardly show that this form of $c_m$ satisfies (10) by substitution.

We then find the form of $v_m$ given this expression for $c_m$. Remember that:

$$v_m = c_m w_m + \frac{1}{N} \sum_{t=m+1}^{M} c_t.$$

We can rearrange (10) to:

$$\frac{1}{N} \sum_{t=m+1}^{M} c_t = 1 - \frac{N - m + 1}{N} c_m$$

from which it follows that:

$$v_m = 1 + c_m \left(w_m - \frac{N - m + 1}{N}\right).$$

Substituting in our expressions for $c_m$ and $w_m$ we thus have:

$$v_m = 1 + \frac{N - M}{(N - m)(N - m + 1)} \left(\frac{1}{q(i_m; i_{1:m-1}, f(\theta_{m-1}), \mathcal{D}_{\text{pool}})} - (N - m + 1)\right)$$

$$v_m = 1 + \frac{N - M}{N - m} \left(\frac{1}{(N - m + 1) \, q(i_m; i_{1:m-1}, \mathcal{D}_{\text{pool}})} - 1\right),$$

which is the form given in the original expression.

To finish our definition, we simply need to derive $C$:

$$
\begin{aligned}
C &= \left( \sum_{m=1}^{M} c_m \right)^{-1} \\
&= \left( N(N-M) \sum_{m=1}^{M} \frac{1}{(N-m)(N-m+1)} \right)^{-1} \\
&= \left( N(N-M) \sum_{m=1}^{M} \frac{1}{N-m} - \frac{1}{N-m+1} \right)^{-1}
\end{aligned}
$$

where we now have a telescopic sum so

$$
\begin{aligned}
&= \left( N(N-M) \left( \frac{1}{N-M} - \frac{1}{N} \right) \right)^{-1} \\
&= \frac{1}{M}.
\end{aligned}
$$

We thus see that our $v_m$ always sum to $M$, giving the quoted form for $\tilde{R}_{\text{LURE}}$ in the main paper.

## B.5 PROOF OF UNBIASEDNESS AND VARIANCE FOR $\tilde{R}_{\text{LURE}}$: THEOREM 3

**Theorem 3.** *$\tilde{R}_{LURE}$ as defined above has the following properties:*

$$
\mathbb{E}\left[ \tilde{R}_{LURE} \right] = r,
$$

$$
\text{Var}\left[ \tilde{R}_{LURE} \right] = \frac{\text{Var}\left[ \mathcal{L}(\mathbf{y}, f_\theta(\mathbf{x})) \right]}{N} + \frac{1}{M^2} \sum_{m=1}^{M} c_m^2 \mathbb{E}_{\mathcal{D}_{\text{pool}}, i_{1:m-1}} \left[ \text{Var}\left[ w_m \mathcal{L}_{i_m} | i_{1:m-1}, \mathcal{D}_{\text{pool}} \right] \right]. \quad (6)
$$

*Proof.* $\tilde{R}_{\text{LURE}}$ is, by construction a linear combination of the weights $a_m$. By Lemma 1 each $a_m$ is an unbiased estimator of $r$. So by the linearity of expectation, $\mathbb{E}\left[ \tilde{R}_{\text{LURE}} \right] = r$.

As in Theorem 1, the variance requires a degree of care because the $a_m$ are not independent. Noting that the expectation does not change through the weighting, we analogously have

$$
\text{Var}\left[ \tilde{R}_{\text{LURE}} \right] = \mathbb{E}\left[ \text{Var}\left[ \frac{1}{M} \sum_{m=1}^{M} c_m a_m \Big| \mathcal{D}_{\text{pool}} \right] \right] + \frac{\text{Var}\left[ \mathcal{L}(\mathbf{y}, f_\theta(\mathbf{x})) \right]}{N}.
$$

Similarly, we also have

$$
\begin{aligned}
\text{Var}\left[ \frac{1}{M} \sum_{m=1}^{M} c_m a_m \Big| \mathcal{D}_{\text{pool}} \right] &= \mathbb{E}\left[ \left( \frac{1}{M} \sum_{m=1}^{M} c_m a_m \right)^2 \Big| \mathcal{D}_{\text{pool}} \right] - \hat{R}^2 \\
&= \frac{1}{M^2} \sum_{m=1}^{M} \sum_{k=1}^{M} c_m c_k \mathbb{E}\left[ a_m a_k | \mathcal{D}_{\text{pool}} \right] - \hat{R}^2 \\
&= \frac{1}{M^2} \sum_{m=1}^{M} \sum_{k=1}^{M} c_m c_k \left( \mathbb{E}\left[ a_m a_k | \mathcal{D}_{\text{pool}} \right] - \hat{R}^2 \right) \\
&= \frac{1}{M^2} \sum_{m=1}^{M} \sum_{k=1}^{M} c_m c_k \mathbb{E}\left[ \left( a_m - \hat{R} \right) \left( a_k - \hat{R} \right) \Big| \mathcal{D}_{\text{pool}} \right].
\end{aligned}
$$

Using the result before that

$$
\mathbb{E}\left[ \left( a_m - \hat{R} \right) \left( a_k - \hat{R} \right) \Big| \mathcal{D}_{\text{pool}} \right] = \begin{cases} \mathbb{E}_{i_{1:m-1}} \left[ \text{Var}\left[ w_m \mathcal{L}_{i_m} | i_{1:m-1}, \mathcal{D}_{\text{pool}} \right] | \mathcal{D}_{\text{pool}} \right] & \text{if } m = k \\ 0 & \text{otherwise} \end{cases}
$$

now gives the desired result by straightforward substitution. $\qquad\square$

## B.6 PROOF THAT $\tilde{R}_{\text{LURE}}$ HAS LOWER VARIANCE THAN $\tilde{R}_{\text{PURE}}$ UNDER REASONABLE ASSUMPTIONS: THEOREM 4

Recall from Theorems 1 and 3 that the variances of the $\tilde{R}_{\text{LURE}}$ and $\tilde{R}_{\text{PURE}}$ estimators are

$$\text{Var}\left[\tilde{R}_{\text{PURE}}\right] = \frac{\text{Var}\left[\mathcal{L}(\mathbf{y}, f_\theta(\mathbf{x}))\right]}{N} + \frac{1}{M^2}\sum_{m=1}^{M} E_m \tag{12}$$

$$\text{Var}\left[\tilde{R}_{\text{LURE}}\right] = \frac{\text{Var}\left[\mathcal{L}(\mathbf{y}, f_\theta(\mathbf{x}))\right]}{N} + \frac{1}{M^2}\sum_{m=1}^{M} c_m^2 E_m, \tag{13}$$

where we have used the shorthand $E_m = \mathbb{E}_{\mathcal{D}_{\text{pool}}, i_{1:m-1}}\left[\text{Var}\left[w_m \mathcal{L}_{i_m}|i_{1:m-1}, \mathcal{D}_{\text{pool}}\right]\right]$. Recall also that

$$c_m^2 = \frac{N^2(N-M)^2}{(N-m)^2(N-m+1)^2}.$$

Though the potential for one to use pathologically bad proposals means that it is not possible to show that $\text{Var}\left[\tilde{R}_{\text{LURE}}\right] \leq \text{Var}\left[\tilde{R}_{\text{PURE}}\right]$ universally holds, we can show this result under a relatively weak assumption that ensures our proposal is "sufficiently good."

To formalize this assumption, we first define

$$F_m := \mathbb{E}_{\mathcal{D}_{\text{pool}}, i_{1:m-1}}\left[\text{Var}\left[\frac{w_m}{\mathbb{E}\left[w_m \mid i_{1:m-1}, \mathcal{D}_{\text{pool}}\right]}\mathcal{L}_{i_m}\bigg|i_{i:m-1}, \mathcal{D}_{\text{pool}}\right]\right] = \left(\frac{N-m+1}{N}\right)^{-2} E_m$$

as the weight-normalized expected variance, where the second form comes from the fact that $\mathbb{E}\left[w_m|i_{1:m-1}, \mathcal{D}_{\text{pool}}\right] = (N-m+1)/N$. Our assumption is now that

$$F_m \geq F_{M-m+1} \quad \forall m : 1 \leq m \leq M/2. \tag{14}$$

Note that a sufficient, but not necessary, condition for this to hold is that the $F_m$ do not increase with $m$, i.e. $F_m \geq F_j \quad \forall(m, j) : 1 \leq m \leq j \leq M$. Intuitively, this is is equivalent to saying that the conditional variances of our normalized weighted losses should not increase as we acquire more points. It is, for example, satisfied by a uniform sampling acquisition strategy (for which all $F_m$ are equal). More generally, it should hold in practice for sensible acquisition strategies as a) our proposal should improve on average as we acquire more labels, leading to lower average variance; and b) higher loss points will generally be acquired earlier so the scaling will typically decrease with $m$. In particular, note that $\mathbb{E}\left[w_m \mathcal{L}_{i_m}|i_{1:m-1}, \mathcal{D}_{\text{pool}}\right] < r$ and is monotonically decreasing with $m$ because it omits the already sampled losses (which is why these are added back in when calculating $a_m$).

This assumption is actually stronger than necessary: in practice the result will hold even if $F_m$ increases with $m$ provided the rate of increase is sufficiently small. However, the assumption as stated already holds for a broad range of sensible proposals and fully encapsulating the minimum requirements on $F_m$ is beyond the scope of this paper.

We are now ready to formally state and prove our result. For this, however, it is convenient to first prove the following lemma, which we will invoke multiple times in the main proof.

**Lemma 2.** *If $a, b, M, N \in \mathbb{N}^+$ are positive integers such that, $M < N$ and $a + b \leq M$, then*

$$\frac{(N-a)^2}{N^2} \geq \frac{(N-M)^2}{(N-b)^2}.$$

*Proof.*

$$\frac{(N-a)^2}{N^2} - \frac{(N-M)^2}{(N-b)^2} = \frac{(N-a)^2(N-b)^2 - N^2(N-M)^2}{N^2(N-b)^2}$$

$$= \frac{2N^3(M-a-b) + N^2(a^2+b^2+4ab-M^2) - 2abN(a+b) + a^2b^2}{N^2(N-b)^2}$$

$$= \frac{1}{N^2(N-b)^2}\left(N(2N-M-a-b)+ab\right)\left(N(M-a-b)+ab\right)$$

$\geq 0$ as $2N \geq M + a + b$ and $M \geq a + b$ so all bracketed terms are positive. $\square$

**Theorem 4.** *If Equation* (14) *in Appendix B.6 holds then* $\mathrm{Var}[\tilde{R}_{LURE}] \leq \mathrm{Var}[\tilde{R}_{PURE}]$. *If $M > 1$ and* $\mathbb{E}_{\mathcal{D}_{\mathrm{pool}}}\left[\mathrm{Var}[w_m \mathcal{L}_{i_1}|\mathcal{D}_{\mathrm{pool}}]\right] > 0$ *also hold, then the inequality is strict:* $\mathrm{Var}[\tilde{R}_{LURE}] < \mathrm{Var}[\tilde{R}_{PURE}]$.

*Proof.* We start by subtracting equation (13) from (12) yielding

$$\mathrm{Var}\left[\tilde{R}_{\mathrm{PURE}}\right] - \mathrm{Var}\left[\tilde{R}_{\mathrm{LURE}}\right] = \frac{1}{M}\sum_{m=1}^{M}(1 - c_m^2)E_m$$

$$= \frac{1}{M}\sum_{m=1}^{M}(1 - c_m^2)\left(\frac{N - m + 1}{N}\right)^2 F_m.$$

Assuming, for now, that $M$ is even and $M < N$, we can now group terms into pairs by counting from each end of the sequence (i.e. pairing the $m$-th and $M - m + 1$-th terms) to yield

$$\mathrm{Var}\left[\tilde{R}_{\mathrm{PURE}}\right] - \mathrm{Var}\left[\tilde{R}_{\mathrm{LURE}}\right] = \frac{1}{M}\sum_{m=1}^{M/2} S_m$$

where

$$S_m := \left(1 - c_m^2\right)\frac{(N - m + 1)^2}{N^2}F_m + \left(1 - c_{M-m+1}^2\right)\frac{(N - M + m)^2}{N^2}F_{M-m+1}$$

$$= \left(\frac{(N - m + 1)^2}{N^2} - \frac{(N - M)^2}{(N - m)^2}\right)F_m + \left(\frac{(N - M + m)^2}{N^2} - \frac{(N - M)^2}{(N - M + m - 1)^2}\right)F_{M-m+1}.$$

We will now show that $S_m \geq 0$, $\forall 1 \leq m \leq M/2$, from which we can directly conclude that $\mathrm{Var}\left[\tilde{R}_{\mathrm{PURE}}\right] \geq \mathrm{Var}\left[\tilde{R}_{\mathrm{LURE}}\right]$. For this, note that $F_m$ and $F_{M-m+1}$ are themselves non–negative by construction.

Consider first the case where $(N - M + m)^2/N^2 \geq (N - M)^N/(N - M + m - 1)^2$. Here the second term in $S_m$ is non-negative. Furthermore, invoking Lemma (2) with $a = m - 1$ and $b = m$ (noting this satisfies $a + b \leq M$ for all $1 \leq m \leq M/2$ as required) shows that

$$\frac{(N - m + 1)^2}{N^2} - \frac{(N - M)^2}{(N - m)^2} \geq 0$$

and so the first term is also positive. It thus immediately follows that $S_m \geq 0$ in this scenario.

When this does not hold, $(N - M + m)^2/N^2 < (N - M)^N/(N - M + m - 1)^2$ and so the second term in $S_m$ is negative. We now instead invoke our assumption that $F_m \geq F_{M-m+1}$, to yield

$$S_m \geq F_m \left(\frac{(N - m + 1)^2}{N^2} - \frac{(N - M)^2}{(N - m)^2} + \frac{(N - M + m)^2}{N^2} - \frac{(N - M)^2}{(N - M + m - 1)^2}\right). \quad (15)$$

We can now invoke Lemma (2) with $a = m - 1$ and $b = M - m + 1$ to show that

$$\frac{(N - m + 1)^2}{N^2} - \frac{(N - M)^2}{(N - M + m - 1)^2} \geq 0$$

and again with $a = M - m$ and $b = m$ to show that

$$\frac{(N - M + m)^2}{N^2} - \frac{(N - M)^2}{(N - m)^2} \geq 0.$$

Substituting these back into (15) thus again yield the desired result that $S_m \geq 0$ as required.

To cover the case where $M$ is odd, we simply need to note that this adds the following additional term as follows:

$$\mathrm{Var}\left[\tilde{R}_{\mathrm{PURE}}\right] - \mathrm{Var}\left[\tilde{R}_{\mathrm{LURE}}\right] = \left(\frac{(N - M/2 + 1/2)^2}{N^2} - \frac{(N - M)^2}{(N - M/2 - 1/2)^2}\right)F_m + \frac{1}{M}\sum_{m=1}^{(M-1/2)} S_m$$

and we can again invoke Lemma (2) with $a = M/2 - 1/2$ and $b = M/2 + 1/2$ to show that this additional term is non–negative.

To cover the case where $M = N$, we simply note that here

$$S_m = \frac{(N - m + 1)^2}{N^2} F_m + \frac{m^2}{N^2} F_{M-m+1}$$

where both terms are clearly positive.

We have now shown that $S_m \geq 0$ in all possible scenarios given our assumption on $F_m$, and so we can conclude that $\text{Var}\left[\tilde{R}_{\text{PURE}}\right] \geq \text{Var}\left[\tilde{R}_{\text{LURE}}\right]$.

Finally, we need to show the inequality is strict if $E_1 > 0$ and $M > 1$. For this we first note that $E_1 > 0$ ensures $F_1 > 0$ and then consider $S_1$ as follows:

$$S_1 = \left(1 - \frac{(N - M)^2}{(N - 1)^2}\right) F_1 + \left(\frac{(N - M + 1)^2}{N^2} - 1\right) F_M$$

and as the second term is clearly negative and $F_1 \geq F_M$,

$$\geq \left(\frac{(N - M + 1)^2}{N^2} - \frac{(N - M)^2}{(N - 1)^2}\right) F_1$$

$$= \frac{(M - 1)(2N^2 - 2MN + M - 1)}{N^2(N - 1)^2} F_1$$

$$> 0$$

as $M > 1$ and $N \geq M$ ensures that each bracketed term is strictly positive. Now as $S_1 > 0$ and $S_m \geq 0$ for all other $m$, we can conclude that the sum of the $S_m$ is strictly positive, and thus that the inequality in strict. $\qquad\square$

## B.7 PROOF OF THE CONSISTENCY OF $\tilde{R}_{\text{LURE}}$: THEOREM 5

**Theorem 5.** *Under the same assumptions as Theorem 2:* $\lim_{M \to \infty} \mathbb{E}\left[\left(\tilde{R}_{LURE} - r\right)^2\right] = 0$.

*Proof.* As before, since $\tilde{R}_{\text{LURE}}$ is unbiased the MSE is simply the variance so:

$$\mathbb{E}\left[\left(\tilde{R}_{\text{PURE}} - r\right)^2\right] = \text{Var}\left[\tilde{R}_{\text{PURE}}\right]$$

$$= \frac{\text{Var}\left[\mathcal{L}(\mathbf{y}, f_\theta(\mathbf{x}))\right]}{N} + \frac{1}{M^2} \sum_{m=1}^{M} c_m^2 \mathbb{E}_{\mathcal{D}_{\text{pool}}, i_{1:m-1}}\left[\text{Var}\left[w_m \mathcal{L}_{i_m} | i_{1:m-1}, \mathcal{D}_{\text{pool}}\right]\right].$$

Taking $N = \alpha M$, we already showed in the proof of Theorem 2 that the first of these terms tends to zero as $M \to \infty$.

We also showed that $\mathbb{E}_{\mathcal{D}_{\text{pool}}, i_{1:m-1}}\left[\text{Var}_{q(i_m; i_{1:m-1}, \mathcal{D}_{\text{pool}})}\left[w_m \mathcal{L}_{i_m}\right]\right]$ is finite given our assumptions. As such, there must be some finite constant $d$ such that

$$\mathbb{E}_{\mathcal{D}_{\text{pool}}, i_{1:m-1}}\left[\text{Var}_{q(i_m; i_{1:m-1}, \mathcal{D}_{\text{pool}})}\left[w_m \mathcal{L}_{i_m}\right]\right] < d$$

and thus

$$\frac{1}{M^2} \sum_{m=1}^{M} c_m^2 \mathbb{E}_{\mathcal{D}_{\text{pool}}, i_{1:m-1}}\left[\text{Var}\left[w_m \mathcal{L}_{i_m} | i_{1:m-1}, \mathcal{D}_{\text{pool}}\right]\right] < \frac{d}{M^2} \sum_{m=1}^{M} c_m^2$$

$$= \frac{d}{M^2} \sum_{m=1}^{M} \left(\frac{N(N - M)}{(N - m)(N - m + 1)}\right)^2$$

and by substituting $N = \alpha M$

$$= \frac{d}{M^2} \sum_{m=1}^{M} \left( \frac{\alpha M (\alpha M - M)}{(\alpha M - m)(\alpha M - m + 1)} \right)^2$$

$$< \frac{d}{M^2} \sum_{m=1}^{M} \left( \frac{\alpha M}{\alpha M - M + 1} \right)^2$$

$$= \frac{d \alpha^2 M}{((\alpha - 1)M + 1)^2}$$

which clearly tends to zero as $M \to \infty$ (remembering that $\alpha > 1$) and we are done. $\qquad \square$

### B.8 PROOF OF BIAS AND VARIANCE OF STANDARD ACTIVE LEARNING ESTIMATOR: THEOREM 6

**Theorem 6.** *Let* $\mu_m := \mathbb{E}\left[\mathcal{L}_{i_m}\right]$ *and* $\mu_{m|i,\mathcal{D}} := \mathbb{E}\left[\mathcal{L}_{i_m}|i_{1:m-1}, \mathcal{D}_{\text{pool}}\right]$. *For* $\tilde{R}$ *(defined in* (1)*):*

$$\mathbb{E}\left[\tilde{R}\right] = \frac{1}{M} \sum_{m=1}^{M} \mu_m \quad (\neq r \text{ in general})$$

$$\text{Var}[\tilde{R}] = \overbrace{\text{Var}_{\mathcal{D}_{\text{pool}}}\left[\mathbb{E}\left[\tilde{R}\middle|\mathcal{D}_{\text{pool}}\right]\right]}^{①} + \overbrace{\frac{1}{M^2} \sum_{m=1}^{M} \mathbb{E}_{\mathcal{D}_{\text{pool}},i_{1:m-1}}\left[\text{Var}\left[\mathcal{L}_{i_m}|i_{1:m-1}, \mathcal{D}_{\text{pool}}\right]\right]}^{②}$$

$$+ \frac{1}{M^2} \sum_{m=1}^{M} \underbrace{\mathbb{E}_{\mathcal{D}_{\text{pool}}}\left[\text{Var}\left[\mu_{m|i,\mathcal{D}}\middle|\mathcal{D}_{\text{pool}}\right]\right]}_{③} + \underbrace{2\,\mathbb{E}_{\mathcal{D}_{\text{pool}}}\left[\text{Cov}\left[\mathcal{L}_{i_m}, \sum_{k<m} \mathcal{L}_{i_k}\middle|\mathcal{D}_{\text{pool}}\right]\right]}_{④}. \tag{7}$$

*Proof.* The result of the bias follows immediately from definition of $\mu_m$ and the linearity of expectations.

For the variance, we have

$$\text{Var}[\tilde{R}] = \mathbb{E}\left[\tilde{R}^2\right] - \left(\mathbb{E}\left[\tilde{R}\right]\right)^2$$

$$= \mathbb{E}_{\mathcal{D}_{\text{pool}}}\left[\mathbb{E}\left[\tilde{R}^2\middle|\mathcal{D}_{\text{pool}}\right]\right] - \left(\mathbb{E}\left[\tilde{R}\right]\right)^2$$

$$= \mathbb{E}_{\mathcal{D}_{\text{pool}}}\left[\text{Var}\left[\tilde{R}\middle|\mathcal{D}_{\text{pool}}\right] + \left(\mathbb{E}\left[\tilde{R}\middle|\mathcal{D}_{\text{pool}}\right]\right)^2\right] - \left(\mathbb{E}\left[\tilde{R}\right]\right)^2$$

$$= \text{Var}_{\mathcal{D}_{\text{pool}}}\left[\mathbb{E}\left[\tilde{R}\middle|\mathcal{D}_{\text{pool}}\right]\right] + \mathbb{E}_{\mathcal{D}_{\text{pool}}}\left[\text{Var}\left[\tilde{R}\middle|\mathcal{D}_{\text{pool}}\right]\right] \tag{16}$$

where the first term is ① from the result. For the second term, introducing the notations $\mu_{|\mathcal{D}} = \mathbb{E}\left[\tilde{R}\middle|\mathcal{D}_{\text{pool}}\right]$ and $\mu_{m|\mathcal{D}} = \mathbb{E}\left[\mathcal{L}_{i_m}|\mathcal{D}_{\text{pool}}\right]$ we have

$$\text{Var}\left[\tilde{R}\middle|\mathcal{D}_{\text{pool}}\right] = \mathbb{E}\left[\left(\frac{1}{M}\sum_{m=1}^{M}\mathcal{L}_{i_m} - \mu_{|\mathcal{D}}\right)^2\middle|\mathcal{D}_{\text{pool}}\right]$$

$$= \frac{1}{M^2}\sum_{m=1}^{M}\sum_{k=1}^{M}\mathbb{E}\left[\left(\mathcal{L}_{i_m} - \mu_{|\mathcal{D}}\right)\left(\mathcal{L}_{i_k} - \mu_{|\mathcal{D}}\right)\middle|\mathcal{D}_{\text{pool}}\right]$$

$$= \frac{1}{M^2}\sum_{m=1}^{M}\sum_{k=1}^{M}\mathbb{E}\left[\left(\mathcal{L}_{i_m} - \mu_{m|\mathcal{D}} + \mu_{m|\mathcal{D}} - \mu_{|\mathcal{D}}\right)\left(\mathcal{L}_{i_k} - \mu_{k|\mathcal{D}} + \mu_{k|\mathcal{D}} - \mu_{|\mathcal{D}}\right)\middle|\mathcal{D}_{\text{pool}}\right],$$

multiplying out terms and using the symmetry of $m$ and $k$

$$= \frac{1}{M^2} \sum_{m=1}^{M} \sum_{k=1}^{M} \mathbb{E}\left[(\mathcal{L}_{i_m} - \mu_{m|\mathcal{D}})(\mathcal{L}_{i_k} - \mu_{k|\mathcal{D}})\big|\mathcal{D}_{\text{pool}}\right]$$

$$+ \frac{2}{M} \sum_{m=1}^{M} \mathbb{E}\left[(\mathcal{L}_{i_m} - \mu_{m|\mathcal{D}})\left(\frac{1}{M}\sum_{k=1}^{M}\mu_{k|\mathcal{D}} - \mu_{|\mathcal{D}}\right)\bigg|\mathcal{D}_{\text{pool}}\right]$$

$$+ \frac{1}{M} \sum_{m=1}^{M} (\mu_{m|\mathcal{D}} - \mu_{|\mathcal{D}})\left(\frac{1}{M}\sum_{k=1}^{M}\mu_{k|\mathcal{D}} - \mu_{|\mathcal{D}}\right)$$

where we have exploited symmetries in the indices. Now, as $\frac{1}{M}\sum_{k=1}^{M}\mu_{k|\mathcal{D}} = \mu_{|\mathcal{D}}$, the second and third terms are simply zero, so we have

$$= \frac{1}{M^2} \sum_{m=1}^{M} \sum_{k=1}^{M} \mathbb{E}\left[(\mathcal{L}_{i_m} - \mu_{m|\mathcal{D}})(\mathcal{L}_{i_k} - \mu_{k|\mathcal{D}})\big|\mathcal{D}_{\text{pool}}\right],$$

separating out the $m = k$ and $m < k$ terms, with symmetry

$$= \frac{1}{M^2} \sum_{m=1}^{M} \text{Var}\left[\mathcal{L}_{i_m}|\mathcal{D}_{\text{pool}}\right] + 2\sum_{k<m} \mathbb{E}\left[(\mathcal{L}_{i_m} - \mu_{m|\mathcal{D}})(\mathcal{L}_{i_k} - \mu_{k|\mathcal{D}})\big|\mathcal{D}_{\text{pool}}\right]$$

$$= \frac{1}{M^2} \sum_{m=1}^{M} \text{Var}\left[\mathcal{L}_{i_m}|\mathcal{D}_{\text{pool}}\right] + 2\mathbb{E}\left[(\mathcal{L}_{i_m} - \mu_{m|\mathcal{D}})\left(\sum_{k<m}(\mathcal{L}_{i_k} - \mu_{k|\mathcal{D}})\right)\bigg|\mathcal{D}_{\text{pool}}\right]$$

$$= \frac{1}{M^2} \sum_{m=1}^{M} \text{Var}\left[\mathcal{L}_{i_m}|\mathcal{D}_{\text{pool}}\right] + 2\,\text{Cov}\left[\mathcal{L}_{i_m}, \sum_{k<m}\mathcal{L}_{i_k}\bigg|\mathcal{D}_{\text{pool}}\right]. \tag{17}$$

Here the second term will yield ④ in the result when substituted back into (16). For the first term, we have by analogous arguments as those used at the start of the proof for $\text{Var}[\tilde{R}]$,

$$\text{Var}\left[\mathcal{L}_{i_m}|\mathcal{D}_{\text{pool}}\right] = \mathbb{E}_{i_{1:m-1}}\left[\text{Var}\left[\mathcal{L}_{i_m}|i_{1:m-1}, \mathcal{D}_{\text{pool}}\right]|\mathcal{D}_{\text{pool}}\right] + \text{Var}\left[\mu_{m|i,\mathcal{D}}|\mathcal{D}_{\text{pool}}\right] \tag{18}$$

where $\mu_{m|i,\mathcal{D}} := \mathbb{E}\left[\mathcal{L}_{i_m}|i_{1:m-1}, \mathcal{D}_{\text{pool}}\right]$ as per the definition in the theorem itself. Substituting this back into (17) and then (16) in turn now yields the desired result through the tower property of expectations, with the first term in (18) producing ② and the second term producing ③. □

## B.9 Proof of Optimal Proposal Distribution: Theorem 7

**Theorem 7.** *Given a non-negative loss, the optimal proposal distribution*

$$q^*(i_m; i_{1:m-1}, \mathcal{D}_{\text{pool}}) = \mathcal{L}_{i_m}/\Sigma_{n\notin i_{1:m-1}}\mathcal{L}_n$$

*yields estimators exactly equal to the pool risk, that is $\tilde{R}_{PURE} = \tilde{R}_{LURE} = \hat{R}$ almost surely $\forall M$.*

*Proof.* We start by proving the result for the simpler case of $\tilde{R}_{\text{PURE}}$ before considering $\tilde{R}_{\text{LURE}}$. To make the notation simpler, we will introduce hypothetical indices $i_t$ for $t > M$, noting that their exact values will not change the proof provided that they are all distinct to each other and the real indices (i.e. that they are a possible realization of the active sampling process in the setting $M = N$).

For $\tilde{R}_{\text{PURE}}$, the proof follows straightforwardly from substituting the definition of the optimal proposal into the $a_m$ form of the estimator

$$
\begin{aligned}
\tilde{R}_{\text{PURE}} &= \frac{1}{M} \sum_{m=1}^{M} a_m \\
&= \frac{1}{M} \sum_{m=1}^{M} \left( w_m \mathcal{L}_{i_m} + \frac{1}{N} \sum_{t=1}^{m-1} \mathcal{L}_{i_t} \right) \\
&= \frac{1}{M} \sum_{m=1}^{M} \left( \frac{1}{N} \sum_{t=m}^{N} \mathcal{L}_{i_t} \right) + \left( \frac{1}{N} \sum_{t=1}^{m-1} \mathcal{L}_{i_t} \right) \\
&= \frac{1}{N} \sum_{t=1}^{N} \mathcal{L}_{i_t},
\end{aligned}
$$

and because all possible indices are uniquely visited

$$
\begin{aligned}
&= \frac{1}{N} \sum_{n=1}^{N} \mathcal{L}_n \\
&= \hat{R}.
\end{aligned}
$$

The proof proceeds identically for the case of $\nabla \mathcal{L}_{i_m}$ because the gradient passes through the summations.

For $\tilde{R}_{\text{LURE}}$, we similarly substitute the optimal proposal into the definition of the estimator

$$
\begin{aligned}
\tilde{R}_{\text{LURE}} &= \frac{1}{M} \sum_{m=1}^{M} v_m \mathcal{L}_{i_m} \\
&= \frac{1}{M} \sum_{m=1}^{M} \mathcal{L}_{i_m} + \frac{N-M}{N-m} \left( \frac{\mathcal{L}_{i_m}}{(N-m+1) q^*(i_m; i_{1:m-1}, f_{\theta_{m-1}}, \mathcal{D}_{\text{pool}})} - \mathcal{L}_{i_m} \right) \\
&= \frac{1}{M} \sum_{m=1}^{M} \mathcal{L}_{i_m} + \frac{N-M}{N-m} \left( \frac{\sum_{t=m}^{N} \mathcal{L}_{i_t}}{(N-m+1)} - \mathcal{L}_{i_m} \right),
\end{aligned}
$$

pulling out the loss

$$
\begin{aligned}
&= \frac{1}{M} \sum_{m=1}^{M} \mathcal{L}_{i_m} \left( 1 - \frac{N-M}{N-m} + (N-M) \underbrace{\sum_{k=1}^{m} \frac{1}{(N-k)(N-k+1)}}_{=m/(N(N-m))} \right) \\
&\quad + \frac{1}{M} \sum_{t=M+1}^{N} \mathcal{L}_{i_t} (N-M) \underbrace{\sum_{k=1}^{M} \frac{1}{(N-k)(N-k+1)}}_{=M/(N(N-M))},
\end{aligned}
$$

simplifying and rearranging

$$
= \frac{1}{M} \sum_{m=1}^{M} \mathcal{L}_{i_m} \left( 1 - \frac{N-M}{N-m} + \frac{N-M}{N-m} \frac{m}{N} \right) + \frac{1}{N} \sum_{t=M+1}^{N} \mathcal{L}_{i_t}
$$

$$
= \frac{1}{M} \sum_{m=1}^{M} \mathcal{L}_{i_m} \left( 1 - \frac{N-M}{N-m} \left( \frac{N-m}{N} \right) \right) + \frac{1}{N} \sum_{t=M+1}^{N} \mathcal{L}_{i_t}
$$

$$
= \frac{1}{N} \sum_{m=1}^{M} \mathcal{L}_{i_m} + \frac{1}{N} \sum_{t=M+1}^{N} \mathcal{L}_{i_t}
$$

$$
= \frac{1}{N} \sum_{n=1}^{N} \mathcal{L}_{n}
$$

$= \hat{R}$ as required. $\qquad \square$

**Remark 2.** *The optimal proposal for estimating the gradient of the pool risk, $\nabla_\phi \hat{R}$, with respect to some scalar $\phi$ is instead*[1]

$$
q^{**}(i_m; i_{1:m-1}, \mathcal{D}_{\text{pool}}) = |\nabla_\phi \mathcal{L}_{i_m}| / \sum_{n \notin i_{1:m-1}} |\nabla_\phi \mathcal{L}_n|.
$$

*Note that when taking gradients with respect to multiple variables, the optimal proposal for each will be different for each.*

---

[1] One can, in principle, actually construct an exact estimator in this scenario as well with the TABI approach of Rainforth et al. (2020) by employing two separate proposals that target $\max(\nabla_\theta \hat{R}, 0)$ and $-\min(\nabla_\theta \hat{R}, 0)$ respectively, then taking the difference between the two resultant estimators.

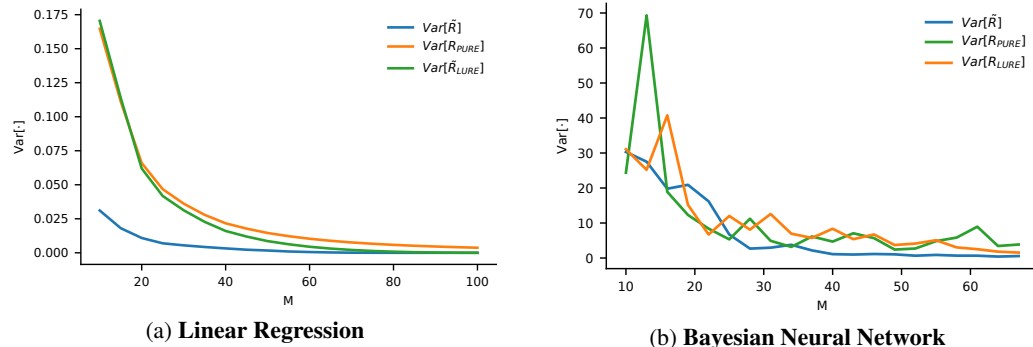

(a) **Linear Regression**  (b) **Bayesian Neural Network**

Figure 5: For linear regression (a) the biased estimator has the lowest variance, and $\tilde{R}_{\text{LURE}}$ improves on $\tilde{R}_{\text{PURE}}$. (b) But for the BNN the variances are more comparable, with $\tilde{R}_{\text{LURE}}$ the lowest.

## C    EXPERIMENTAL DETAILS

### C.1    LINEAR REGRESSION

Our training dataset contains a small cluster of points near $x = -1$ and two larger clusters at $0 \leq x \leq 0.5$ and $1 \leq x \leq 1.5$, sampled proportionately to the 'true' data distribution. The data distribution from which we select data in a Rao-Blackwellised manner has a probability density function over x equal to:

$$P(\text{x} = X) = \begin{cases} 0.12 & -1.2 \leq \text{x} \leq -0.8 \\ 0.95 & 0.0 \leq \text{x} \leq 0.5 \\ 0.95 & 1.0 \leq \text{x} \leq 1.5 \end{cases}$$

while the distribution over y is then induced by:

$$\text{y} = \max(0, x) \cdot \left( |x|^{\frac{3}{2}} + \frac{\sin(20x)}{4} \right).$$

We set $N = 101$, where there are 5 points in the small cluster and 96 points in each of the other two clusters, and consider $10 \leq M \leq 100$. We actively sample points without replacement using a geometric heuristic that scores the quadratic distance to previously sampled points and then selects points based on a Boltzman distribution with $\beta = 1$ using the normalized scores.

Here, we also show in Figure 6 results that are collected using an epsilon-greedy acquisition proposal. The results are aligned with those from the other acquisition distribution we consider in the main body of the paper. This proposal selects the point that is has the highest total distance to all previously selected points with probability 0.9 and uniformly at random with probability $\epsilon = 0.1$. That is, the acquistion proposal is given by:

$$P(i_m = j; i_{1:m-1}, \mathcal{D}_{\text{pool}}) = \begin{cases} 1 - \epsilon + \frac{\epsilon}{|\mathcal{D}_{\text{pool}}|} & \arg\max_{j \notin \mathcal{D}_{\text{train}}} \sum_{k \in \mathcal{D}_{\text{train}}} |x_k - x_j| \\ \frac{\epsilon}{|\mathcal{D}_{\text{pool}}|} & \text{otherwise} \end{cases}$$

where of course $\mathcal{D}_{\text{train}}$ are the $i_{1:m-1}$ elements of $\mathcal{D}_{\text{pool}}$.

For all graphs we use 1000 trajectories with different random seeds to calculate error bars. Although, of course, each regression and scoring is deterministic, the acquistion distribution is stochastic.

Although the variance of the estimators can be inferred from Figure 2a, we also provide Figure 5a which displays the variance of the estimator directly.

### C.2    BAYESIAN NEURAL NETWORK

We train a Bayesian neural network using variational inference (Jordan et al., 1999). In particular, we use the radial Bayesian neural network approximating distribution (Farquhar et al., 2020). The details of the hyperparameters used for training are provided in Table 1.

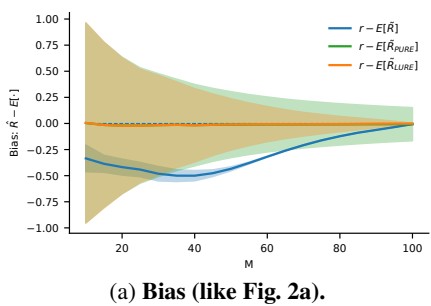

(a) **Bias (like Fig. 2a).**

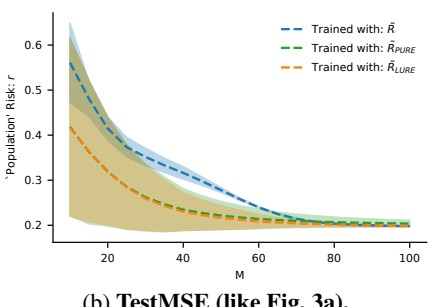

(b) **TestMSE (like Fig. 3a).**

Figure 6: Adopting an alternative proposal distribution—here an epsilon-greedy adaptation of a distance-based measure—does not change the overall picture for linear regression.

| Hyperparameter | Setting description |
|---|---|
| Architecture | Convolutional Neural Network |
| Conv 1 | 1-16 channels, 5x5 kernel, 2x2 max pool |
| Conv 2 | 16-32 channels, 5x5 kernel, 2x2 max pool |
| Fully connected 1 | 128 hidden units |
| Fully connected 2 | 10 hidden units |
| Loss function | Negative log-likelihood |
| Activation | ReLU |
| Approximate Inference Algorithm | Radial BNN Variational Inference (Farquhar et al., 2020) |
| Optimization algorithm | Amsgrad (Reddi et al., 2018) |
| Learning rate | $5 \cdot 10^{-4}$ |
| Batch size | 64 |
| Variational training samples | 8 |
| Variational test samples | 8 |
| Variational acquisition samples | 100 |
| Epochs per acquisition | up to 100 (early stopping patience=20), with 1000 copies of data |
| Starting points | 10 |
| Points per acquistion | 1 |
| Acquisition proposal distribution | $q(i_m; i_{1:m-1}, \mathcal{D}_{\text{pool}}) = \frac{e^{Ts_i}}{\sum e^{Ts_i}}$ |
| Temperature: $T$ | 10,000 |
| Scoring scheme: $s$ | BALD (M.I. between $\theta$ and output distribution) |
| Variational Posterior Initial Mean | He et al. (2016) |
| Variational Posterior Initial Standard Deviation | $\log[1 + e^{-4}]$ |
| Prior | $\mathcal{N}(0, 0.25^2)$ |
| Dataset | Unbalanced MNIST |
| Preprocessing | Normalized mean and std of inputs. |
| Validation Split | 1000 train points for validation |
| Runtime per result | 2-4h |
| Computing Infrastructure | Nvidia RTX 2080 Ti |

Table 1: Experimental Setting—Active MNIST.

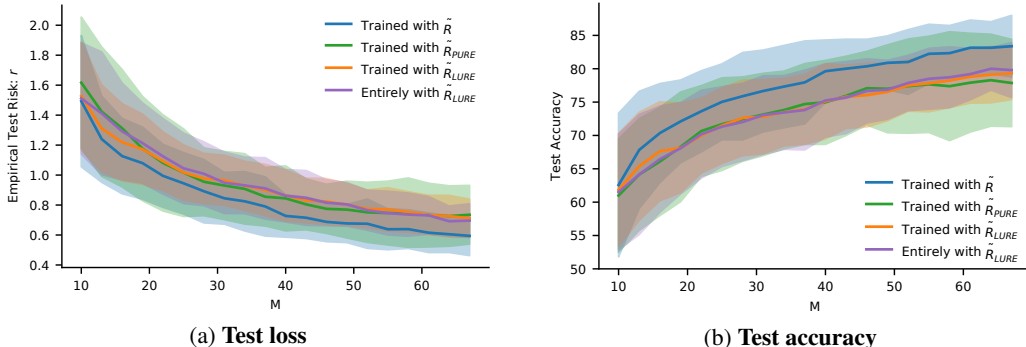

(a) **Test loss**  (b) **Test accuracy**

Figure 7: We contrast the effect of using $\tilde{R}_{\text{LURE}}$ throughout the entire acquisition procedure and training (rather than using the same acquisition procedure based on $\tilde{R}$ for all estimators). The purple test performance and orange are nearly identical, suggesting the result is not sensitive to this choice.

The unbalanced dataset is constructed by first noising 10% of the training labels, which are assigned random labels, and then selecting a subset of the training dataset such that the numbers of examples of each class is proportional to the ratio (1., 0.5, 0.5, 0.2, 0.2, 0.2, 0.1, 0.1, 0.01, 0.01)—that is, there are 100 times as many zeros as nines in the unbalanced dataset. (Figure 3f shows a version of this experiment which uses a balanced dataset instead, in order to make sure that any effects are not entirely caused by this design choice.) In fact, we took only a quarter of this dataset in order to speed up acquisition (since each model must be evaluated many times on each of the candidate datapoints to estimate the mutual information). 1000 validation points were then removed from this pool to allow early stopping. The remaining points were placed in $\mathcal{D}_{\text{pool}}$. We then uniformly selected 10 points from $\mathcal{D}_{\text{pool}}$ to place in $\mathcal{D}_{\text{train}}$. Adding noise to the labels and using an unbalanced dataset is designed to mimic the difficult situations that active learning systems are deployed on in practice, despite the relatively simple dataset. However, we used a simple dataset for a number of reasons. Active learning is very costly because it requires constant retraining, and accurately measuring the properties of estimators generally requires taking large numbers of samples. The combination makes using more complicated datasets expensive. In addition, because our work establishes a lower bound on architecture complexity for which correcting the active learning bias is no longer valuable, establishing that lower bound with MNIST is in fact a stronger result than showing a similar result with a more complex model.

The active learning loop then proceeds by:

1. training the neural network on $\mathcal{D}_{\text{train}}$ using $\tilde{R}$;
2. scoring $\mathcal{D}_{\text{pool}}$;
3. sampling a point to be added to $\mathcal{D}_{\text{train}}$;
4. Every 3 points, we separately trained models on $\mathcal{D}_{\text{train}}$ using $\tilde{R}$, $\tilde{R}_{\text{PURE}}$, and $\tilde{R}_{\text{LURE}}$ and evaluate them.

This ensures that all of the estimators are on data collected under the same sampling distribution for fair comparison. As a sense-check, in Figures 7a and 7b we show an alternate version in which the first step trains with $\tilde{R}_{\text{LURE}}$ instead of $\tilde{R}$, and find that this does not have a significant effect on the results.

When we compute the bias of a fixed neural network in Figure 2b, we train a single neural network on 1000 points. We then sample evaluation points using the acquisition proposal distribution from the test dataset and evaluate the bias using those points.

In Figures 8a and 8b we review the graphs shown in Figures 3b and 3c, this time showing standard errors in order to make clear that the biased $\tilde{R}$ estimator has better performance, while the earlier figures show that the performance is quite variable.

We considered a range of alternative proposal distributions. In addition to the Boltzman distribution which we used, we considered a temperature range between 1,000 and 20,000 finding it had relatively little effect. Higher temperatures correspond to more certainly picking the highest mutual information

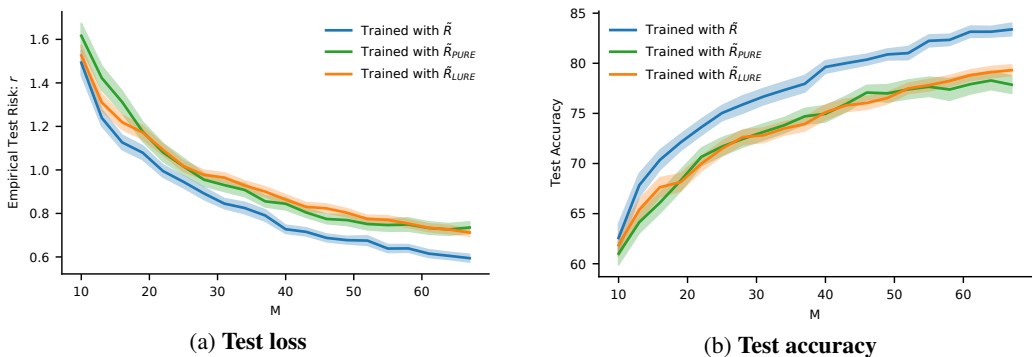

(a) **Test loss**

(b) **Test accuracy**

Figure 8: Versions of Figures 3b and 3c shown with standard errors (45 points) instead of standard deviations. This makes it clearer that the biased $\tilde{R}$ has better performance, even if only marginally so.

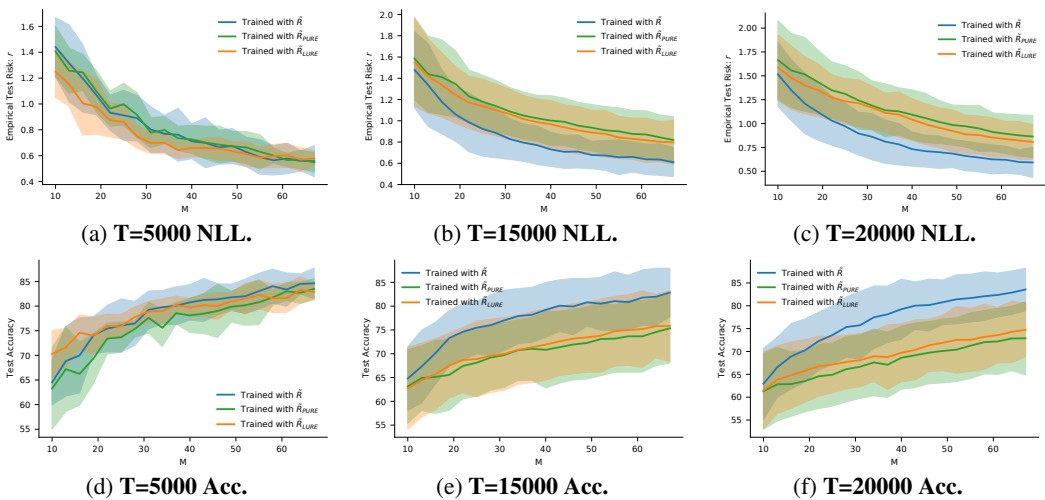

(a) **T=5000 NLL.**

(b) **T=15000 NLL.**

(c) **T=20000 NLL.**

(d) **T=5000 Acc.**

(e) **T=15000 Acc.**

(f) **T=20000 Acc.**

Figure 9: Higher temperatures approach a deterministic acquisition function. These also tend to increase the variance of the risk estimator because the weight associated with unlikely points increases, when it happens to be selected. The overall pattern seems fairly consistent, however.

point, which approaches a deterministic proposal. We found that because the mutual information had to be estimated, and was itself a random variable, different trajectories still picked very different sets of points. However, for very high temperatures the estimators became higher variance, and for lower temperatures, the acquisition distribution became nearly uniform. In Figure 9 we show the results of networks trained with a variety of temperatures other than the 10,000 ultimately used. We also considered a proposal which was simply proportional to the scores, but found this was also too close to sampling uniformly for any of the costs or benefits of active learning to be visible.

We considered Monte Carlo dropout as an alternative approximating distribution (Gal & Ghahramani, 2015) (see Figures 3e and 10b). We found that the mutual information estimates were compressed in a fairly narrow range, consistent with the observation by Osband et al. (2018) that Monte Carlo dropout uncertainties do not necessarily converge unless the dropout probabilities are also optimized (Gal et al., 2017a). While this might be good enough when only the *relative* score is needed in order to calculate the argmax, for our proposal distribution we would ideally prefer to have good *absolute* scores as well. For this reason, we chose the richer approximate posterior distribution instead.

Last, we considered a different architecture, using a full-connected neural network with a single hidden layer with 50 units, also trained as a Radial BNN. This showed higher variance in downstream performance, but was broadly similar to the convolutional architecture (see Figures 10d and 10e).

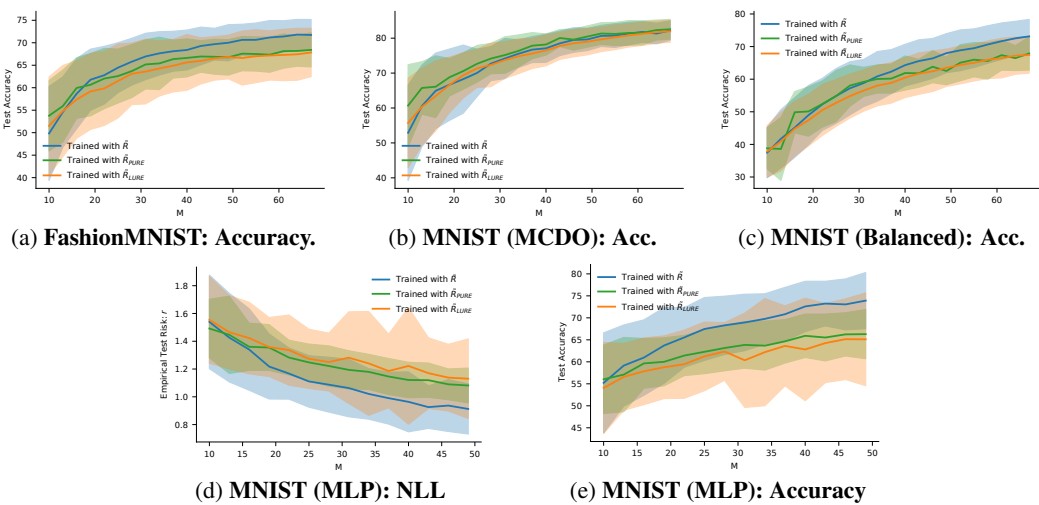

(a) **FashionMNIST: Accuracy.**  (b) **MNIST (MCDO): Acc.**  (c) **MNIST (Balanced): Acc.**

(d) **MNIST (MLP): NLL**  (e) **MNIST (MLP): Accuracy**

Figure 10: Further downstream performance experiments. (a)-(c) are partners to Figures 3d, 3e, and 3f. (d) and (e) show similar results for a smaller multi-layer perceptron (with one hidden layer of 50 units). In all cases the results broadly mirror the results in the main paper.

| Reference | Application | Corrects Bias | Acknowledges Bias | Notes |
|---|---|---|---|---|
| (Sener & Savarese, 2018) | | | | |
| (Shen et al., 2018) | ✓ | | | |
| (Beluch et al., 2018) | | | | |
| (Haut et al., 2018) | ✓ | | | |
| (Sinha et al., 2019) | | | | |
| (Siddhant & Lipton, 2018) | ✓ | | ✓ | |
| (Ghosal et al., 2019) | ✓ | | | |
| (Yang et al., 2018) | ✓ | | | |
| (Yoo & Kweon, 2019) | | | | |
| (Kirsch et al., 2019) | | | | |
| (Huang et al., 2018) | | | ✓ | |
| (Wen et al., 2018) | ✓ | | | |
| (Chen et al., 2019) | ✓ | | | Discusses bias in $\mathcal{D}_{\text{pool}}$. |
| (Zhang & Lee, 2019) | ✓ | | | Discusses bias in $\mathcal{D}_{\text{pool}}$. |
| (Kellenberger et al., 2019) | ✓ | | | |

Table 2: Existing applications of deep active learning rarely acknowledge the bias introduced by actively sampling points and do not, to the best of our knowledge, try to correct it.

## D  DEEP ACTIVE LEARNING IN PRACTICE

In Table 2, we show an informal survey of highly cited papers citing Gal et al. (2017b), which introduced active learning to computer vision using deep convolutional neural networks. Across a range of papers including theory papers as well as applications ranging from agriculture to molecular science only two papers acknowledged the bias introduced by actively sampling and none of the papers took steps to address it. It is worth noting, though, that at least two papers motivated their use of active learning by observing that they expected their training data to already be unrepresentative of the population data and saw active learning as a way to address *that* bias. This does not quite work, unless you explicitly assume that the actively chosen distribution is more like the population distribution, but is an interesting phenomenon to observe in practical applications of active learning.

