# OpenReview forum: "On Statistical Bias In Active Learning: How and When to Fix It"
_ICLR.cc/2021/Conference — ICLR 2021 Spotlight_

### Official Review · AnonReviewer4 · 2020-10-27
**Insightful paper, but with limited experiments**

**Rating:** 7
**Confidence:** 4

**Review:**

**The changes of the review after rebuttal are  indicated in bold text.**

### Summary
The paper discusses the issue of bias in active learning, the problem of actively selecting new training points to label when the labelling task is expensive.
Because of the labelling cost, we wish to select only the most informative points, which are not representative of the full dataset, hence incurring a statistical bias.
The classical estimator of the risk is the unweighted average loss, which carries the bias of the active training set.
The authors propose 2 unbiased estimators of the risk, named $R_{PURE}$ and $R_{SURE}$, in order to treat the statistical bias.
Although both unbiased, they differ in the ponderation on the loss for each active sample and do not have the same variance.
While $R_{PURE}$ depends on the order in which the observations are actively chosen, $R_{SURE}$ removes this dependency.
Both estimators are tested and compared with the classical biased risk estimator in two settings: a linear model applied on a toy dataset,
and a Bayesian convolutional neural network (BNN) applied on a modified version of MNIST putting the data in more difficult setting.
The authors show that both their estimators behave much better than the standard one in the linear case, while they are very similar albeit a little worse in the MNIST problem.
They finish the paper by explaining that unbiased estimators of the risk are better for low parametrized models, such as the linear one, which are very dependent of the statistical bias;
while they decrease a bit the performances of an over-parametrized estimator, such as the BNN, for which the statistical bias seems to be cancelled out with the overfitting bias.

### Strengths
The paper is well written and clear. It addresses the interesting problem of active learning, and it is very insightful of the particular issue of bias in that context.
Each expression is explained with the meaning of its components, and the authors attempt to provide insight to both the positive and negative results.
It is very refreshing to see negative results, showing clearly the limitations of the proposed estimators, and to have a clear suggestion from the authors as to when to use which estimator (theirs or the biased one).
The math seems correct, although I have not checked all the proofs.

### Weaknesses and concerns
1. Proofs
The explanations often rely more on intuition than on mathematical proof. While they may be true, it seems to me that it lacks a bit of theoretical grounding.
**After discussion with the authors, it turns out that this comment was not clear enough. It concerned only the paragraphs after Theorems 3 and 5. I apreciate greatly that the authors made an effort to clarify these paragraphs.**

2. Experiment
    a. Limitations
The experimental study is limited: the estimators have been applied only on two extreme models (a linear model and a BNN), and for each case, only one dataset has been tested. While both are very much used in practice, it is not enough to draw conclusions. There should at least be an emphasis of that, saying that it is only a preliminary, and that more tests will be performed in a more extensive version of the work. Has it been tested on more models (e.g. random forests) or on various architectures of neural networks, as well as on several datasets?
As for the choice of the proposal distribution, it seems only one was tested in the linear setting, while a range of them were tested for the BNN. This should be more explicit in the main article, referring to the appendix for more details.
    b. Reproducibility and relevance of the experiment
There are issues on the toy example:
      * it is a highly nonlinear dataset on which a linear model is trained, which is weird: what happens when using the proposed estimators to train a linear model on (close to) linear data?
      * the density function on which $X$ is drawn does not seem to sum to 1.
      * there seems to be an issue in the generation of $y$ (Eq. 124), as there no bias added, yet Figure 1 shows an oscillation for $y$ between 1 and 2.5 and $x$ in $[1, 1.5]$, while I found an oscillation around 0 by applying the formula as specified

  In the modified MNIST dataset:
      - it seems to me that the proportion of each class used for the pool subset makes this particular dataset somewhat easier rather than harder, as it basically removes the ambiguity between numbers that look alike; for instance, there are very few observations of 9, and a few of 6, and there are 5 times less 7 than 1; in such a case, the accuracy is not a good measure of the quality, as it will be good even if the smaller classes are poorly classified.
      - Has there been a test on a balanced subset?

**In the revision, the authors added a few experiments to complete the first ones, namely adding a third dataset (Fashion MNIST), testing a balanced subset for MNIST, and testing another acquisition function to see its impact on the results. I think this gives more grounding to the nice theoretical results and insights.**

3. Active sampling for test
The authors claim that their estimators should have a positive impact on the actively selecting samples at test time. This claim appears in the introduction and in the conclusion, but it seems more like intuition and there is no explanation as to why this may be true.
I suggest removing it from the introduction, as it is clearly future work, and if possible add some explanation behind the intuition.

### Minor comments:
- Please put equation numbers only on equations you refer to, not on all equations.
- The name $R_{SURE}$ can be confused with Stein's Unbiased Risk Estimator (even though I assume it is so as a reference to that estimator).
- In Section 6, the acronym BALD is not defined
- In Figure 4a, the colors and order in the legend are reversed for $R_{PURE}$ and $R_{SURE}$

### Overall evaluation
I believe this is very promising and insightful work, tackling the important issue of bias.
However, I cannot give it a high score due to the issues I raised about the experiment.

**The revision answered my concerns about the experiment by enlarging it, which is why I upgrade the score I gave. I think it is a very good and interesting paper.**

---

> ### Author Response · Authors · 2020-11-18
> **Thank you for your review.**
>
> > Proofs The explanations often rely more on intuition than on mathematical proof. While they may be true, it seems to me that it lacks a bit of theoretical grounding.
>
> We are not quite sure what you are referring to here. Theorems 1 through 6 all have detailed mathematical proofs, provided in the appendix. If you have any concerns that any steps are less than mathematically rigorous, please help us identify these spots so we can improve this.
>
> One design choice we have made is to move the proofs for the clear-cut results to the appendix, for readers who are interested in the mechanism of the proof to explore separately. Meanwhile the more informal arguments are kept in the main body of the paper to build intuition and allow the readers to be in a better position to evaluate their merits and shortcomings as they read. Unfortunately, the long and numerous nature of the proofs and tight space requirements makes it difficult to also provide proof sketches in the main paper. This might give the impression that much of the contribution is not rigorously proven, if going by space taken up in the main paper alone. Perhaps this is what you are noticing? If so, please do consider the extensive mathematically rigorous contribution that is present *in addition* to more heuristic arguments.
>
> > Experiment a. Limitations
>
> We agree entirely that a fuller version of this work, suitable for a journal setting, would need fuller experimental substantiation and will make note of this, but we also emphasize that we have already made substantial theoretical contributions and there are limits to one can be achieved in this format. Nonetheless, to alleviate your concerns we have added experiments based on your sensible suggestions that we believe substantially strengthen the work:
> - An additional dataset, Fashion MNIST.  [Figures 2c, 3d, 4c, and 10a]
> - An additional architecture without convolutions. (We have not considered random forests.)  [Figures 10d and 10e]
> - Monte Carlo dropout as an alternative inference approximation. [Figures 3e and 10b]
> - An ablation study of proposal distributions for the Bayesian neural network [Figures 9a-f], as well as an additional proposal distribution for the linear model.[Figures 6a and 6b]
>
> All of these variants show broadly similar results, and can be found within what are now sections 7 and 8 as well as the Appendix section C.
>
> >b. Reproducibility and relevance
> > nonlinear dataset on which a linear model is trained
>
> It is crucially important for the linear regression to be on non-linear data. A linear regression on (almost) linear data is insensitive to which points are sampled: any two points define the line. This makes active learning superfluous.
>
> > density function… does not seem to sum to 1.
>
> Apologies, there was a typesetting error in the density which has now been corrected.
>
> > issue in the generation of (Eq. 124)
>
> Apologies, there is a missing addition between the sin term and the $|x|^{3/2}$.  We will further make our code available after publication.
>
> > proportion of each class used… makes this particular dataset somewhat easier rather than harder
>
> The goal of using unbalanced data is that this makes the effect of any bias more severe, not that it makes it more challenging to approximate the function. It was supposed to show a best-case-scenario for the unbiased estimator. But we agree that this is an important thing to make sure is not affecting results, so we have included a version on balanced data [Figures 3f and 10c]. We found that this produces similar behaviour.
>
> In addition, we have added results on FashionMNIST with unbalanced data. In Fashion MNIST the 'confusable pairs’ are less straightforward than MNIST.
>
> > Active sampling for test.
>
> We included this as a potentially exciting area of future work, but agree that we have overemphasised it and gotten ahead of ourselves a little! As noted in the response to AnonReviewer1, the rationale is that at test time the focus is directly on estimating the risk, for which our debiasing is particularly important (whereas there is an extra degree of separation in training as characterized by the overfitting bias). Nonetheless we agree this is overly preliminary for the introduction and for the time being, we will simply point out that after optimization is complete, there is no reason not to use $R_{\text{SURE}}$ rather than $\tilde{R}$ for any risk estimates in typical cases.
>
>
> > Minor comments
>
> Thank you also for the minor comments which we have addressed.
>
> We are brainstorming alternative names, but have not yet come up with something clearer and less ambiguous. We will continue to think about this.

---

> > ### Comment · AnonReviewer4 · 2020-11-23
> > **More details on my comment about intuition/theoretical grounding, and answer to the added experiments**
> >
> > > > Proofs The explanations often rely more on intuition than on mathematical proof. While they may be true, it seems to me that it lacks a bit of theoretical grounding.
> >
> > > We are not quite sure what you are referring to here. Theorems 1 through 6 all have detailed mathematical proofs, provided in the appendix.
> >
> > Sorry for the confusion, I was actually talking about the paragraphs right after Theorems 3 and 5, and not about the theorems and proofs that look good. I really like the fact that you try to explain the theoretical results, but the explanations are not always clear to me. For instance when you say that the variance of $R_{SURE}$ is typically better than that of $R_{PURE}$, and that $c_m$ generally upweight the lower variance latter terms and downweight the higher variance earlier terms, what makes it balance toward 0 in overall? It is not very obvious in the experiments either (apart from Fig. 2 (a)). Also, it seems to me that $1/M\sum_{m=1}^Mc_m $ is not equal to 1, but tends to 1  when $N \gg M$, and that the average value of $w_m$ should depend on the proposal distribution $q$.
> >
> > After Theorem 5, you mention "when things work well", perhaps just putting a semicolon instead of a point at the end of the sentence will make it clearer that you explain it immediately after.  Afterwards, you mention that the term $(5)$ reflects correlations between loss at different iterations, which you estimators have eliminated, but is it really true for $R_{PURE}$, since it gives more weight to earlier samples?
> >
> > > Experiment a. Limitations
> >
> > Thank you for addressing my concerns about the limitations of the experiments, and adding comparisons with another acquisition strategy and another dataset, as well as adding the comparison with balanced MNIST. I believe it is now good enough for a paper that size with a big focus on theory.
> >
> > > We are brainstorming alternative names, but have not yet come up with something clearer and less ambiguous. We will continue to think about this.
> >
> > I understand it is a difficult task... How about $R_{CURE}$ as it cures the issue of $R_{PURE}$ regarding the weights on samples' order (although I don't have ideas as to what the C would stand for)?
> >
> > I also like the new section 4, I think it adds an interesting dicussion.

---

> > > ### Author Response · Authors · 2020-11-23
> > > **Thanks for the follow-up points.**
> > >
> > > Thank you for your very helpful follow-up; it definitely highlights some parts that we can improve the clarity for, while we are delighted that the new experiments and Section 4 have cleaned up some of your big concerns. We go through each of your points below and have uploaded a new revision to help address them in the paper itself.
> > >
> > > Re intuition after Theorem 3
> > > > you say that the variance of $R_{SURE}$ is typically better than that of $R_{PURE}$ and that $c_m$ generally upweight the lower variance latter terms and downweight the higher variance earlier terms, what makes it balance toward 0 in overall?
> > >
> > > Our argument here is admittedly informal and could have been better worded. When thinking about whether the overall balance will be towards reducing variance, we note that $R_{PURE}$ is the estimator with the unusual property of arbitrarily placing extra weight on earlier points; even with a uniform proposal it would still produce uneven weights.  The key high-level intuition is simply that $R_{SURE}$ corrects this pathology, by ensuring the expectation of the weighting $v_m$ is the same for all $m$, and this should intuitively reduce the variance.
> > >
> > > We have rewritten this paragraph to make this clearer and remove some of the more confusing informal arguments.  We are also nearly finished with developing a more formal demonstration that the variance of $R_{SURE}$ is lower than $R_{PURE}$ under some weak assumptions, but unfortunately it is unlikely this will be ready before the end of the discussion period.
> > >
> > > > it seems to me that $\frac{1}{M}\sum_{m=1}^{M}c_m$ is not equal to 1, but tends to 1 when $N≫M$
> > >
> > > It is indeed the case that this holds exactly for all $M$ and $N$.  This result is formally demonstrated in Appendix B.4.
> > > We have added several steps in the proof just before what is now Equation 9 to clarify this derivation.
> > >
> > > >the average value of $w_m$ should depend on the proposal distribution $q$
> > >
> > > This is actually not the case ($E[w_m]=(N-m+1)/N$) because there is a cancelling effect as follows:
> > > $$
> > > \begin{align}
> > > E[w_m] &= E_{q}\left[\frac{1}{Nq}\right]\\\\
> > > &= \sum_{D_{\text{pool}}} \frac{1}{N}\\\\
> > > &= \frac{N-m+1}{N},
> > > \end{align}
> > > $$
> > > remembering that there are $N-m+1$ remaining points in $D_{\text{pool}}$ at each step.
> > >
> > > > "when things work well", perhaps just putting a semicolon instead of a point at the end of the sentence will make it clearer that you explain it immediately after.
> > >
> > > We agree this could have been clearer.  In response to another reviewer’s comments also, we have rewritten these sentences to remove the unclarity of “when things are working well”.
> > >
> > > > Afterwards, you mention that the term (5) reflects correlations between loss at different iterations, which you estimators have eliminated, but is it really true for $R_{PURE}$, since it gives more weight to earlier samples?
> > >
> > > We have edited the text to clarify exactly what is going on in this claim. The analogy of the (5) term for $R_{PURE}$ is $\text{Cov}(a_m, a_k)$.  Though we agree that this is not immediately intuitive, in the proof of Theorem 1 we do formally demonstrate that $\text{Cov}(a_m, a_k) = E[(a_m - r)(a_k -r)] = 0$ for $k \neq m$. This also then carries over to $R_{SURE}$ as the relevant components are just scaled by a constant term: $c_m a_m$. Specifically, we have $\text{Cov}(c_m a_m, c_k a_k) = c_m c_k \text{Cov}(a_m, a_k) = 0$ since $c_m$ and $c_k$ are not random variables.
> > >
> > > > Alternative name
> > >
> > > $R_\text{CURE}$ is a good suggestion, and we think we can make this work. However, to avoid confusion at this stage we propose that we make this change in the document after the discussion period has ended.

---

### Official Review · AnonReviewer3 · 2020-10-28
**Interesting take on estimating expected risk from adaptively acquired data, but deficient in theory and experiments**

**Rating:** 4
**Confidence:** 4

**Review:**

**Summary**:
The authors consider the bias (in the risk) introduced by active sampling strategies with respect to the true underlying data generating distribution. They then propose two estimators of the risk -- SURE and PURE -- that are unbiased and asymptotically consistent under certain assumptions. The authors consider two toy examples where they show the existence of active learning bias, and the effect of removing this. In particular, the authors notice that while training a linear regression (based on actively acquired datapoints) with their unbiased risks improves test loss, the opposite happens for a Bayesian Neural Network (BNN). They then provide a potential explanation for this phenomenon.

**+ves**:
+ The risk estimator proposals make sense, and their motivation is clear.
+ The calculations (and the proofs) are presented very clearly; the overall writing quality and the story arc of the paper is very clear

**Concerns**
- The condition that the proposal distribution needs a non-trivial weight on the entire pool of data seems too strong. Intuitively, if there are irrelevant parts of the distribution (say, far away from the decision boundary), a proposal distribution that doesn't focus there shouldn't introduce too much bias. It would have been nice to see a more nuanced version of this analysis.
- In fact, the variance depends on this weight (as $1/\beta^2$). This could imply an extremely large variance in several reasonable scenarios, and hence affect the performance of any active learning algorithm.
- The authors address the above concern by claiming that this paper is only about the "statistical bias" of active learning. I find this unsatisfactory since without the aspect of downstream learning, this is simply an estimation of an expectation with samples from a mismatched distribution.
- The experimental section could have used significant strengthening. Especially given the difficulty of making theoretical claims about generalization, it would have been useful to see more experiments that shed light on how a learning algorithm that adaptively selects data can benefit from the proposal of this paper.

---

> ### Author Response · Authors · 2020-11-18
> **Thank you for your review. Part 1.**
>
> > The condition that the proposal distribution needs a non-trivial weight on the entire pool of data seems too strong. Intuitively, if there are irrelevant parts of the distribution (say, far away from the decision boundary), a proposal distribution that doesn't focus there shouldn't introduce too much bias.
>
> This raises an interesting issue and we have added discussion of this point to a new section 4.
>
> Our approach does actually allow proposals with zero mass on some datapoints if one is willing to accept a small amount of bias.  We can characterize this bias exactly as follows, where we use $I$ to denote the set of points we ignore by placing zero mass on them in our proposal and $\tilde{R}^I_{\text{SURE}}$ to denote the resulting estimator:
> $$
> \\begin{align}
> \\mathbb{E}[\\tilde{R}^I_{\\text{SURE}}] &= \\mathbb{E}\left[\\mathbb{E}[R_{\\text{SURE}}|D_{\text{pool}}] - \mathbb{E}\left[\frac{1}{N} \sum_{n \in I} L_n|D_{\text{pool}}\right]\right] \\\\
> &= r-\mathbb{E}\left[\frac{1}{N}\sum_{n\in I} L_n\right]
> \end{align}
> $$
> such that the bias is $-\mathbb{E}\left[\frac{1}{N}\sum_{n\in I} L_n\right]$. This bias will never, in general, be zero as we can never know the class/output of an unseen point for certain without labelling, even if it is far from the decision boundary. Our philosophy has been agnostic over possible acquisition strategies, and to describe the conditions required for an unbiased risk estimate under acquisition.
>
> But as you say, it may be the case that the bias introduced is sometimes small and provides advantages in the acquisition, such that it is worth taking such a strategy. However, good acquisitions proposals should account for these points having small expected losses (and thus little influence on training) by placing appropriately low proposal mass on them.
>
> > In fact, the variance depends on this weight (as $1/\beta^2$). This could imply an extremely large variance in several reasonable scenarios, and hence affect the performance of any active learning algorithm
>
> This concern seems to be based on a confusion: the variance does not vary as $1/\beta^2$. We introduce the constant $\beta$ as an analytical tool to prove the asymptotic convergence of the estimators. The bound being introduced in the proof is extremely loose and only being used to show that the variance is finite, rather than provide any accurate approximation.  $\frac{1}{\beta}$ corresponds to the index with the largest weight and we are using it to bound the variance of all the other terms.  However, the index with the largest weight is also the least likely to be selected and this then cancels things out in practice; the variance of the largest weight is not representative of the average of the variances of the weights (which is what we care about).  This kind of logic is relatively common in the importance sampling literature (e.g. http://www.numdam.org/article/PS_2007__11__427_0.pdf) and does not imply that the variance will be large.  Note also that importance sampling (upon which our approach is based) was originally introduced to the Monte Carlo literature as a mechanism for variance reduction: the variance is only large if we use a proposal that is worse than selecting datapoints at random.
>
> In fact, we show a number of positive results about the variance.  Firstly we characterize it exactly in Theorem 1 and 3 where we see there is no dependence on $\beta$.  Secondly, in Theorem 6 we show that the optimal proposal produces *zero* variance with our estimator.  Finally, in Figure 5 in the Appendix we show that the variance of our estimators with real example acquisitions strategies produces variances that are very similar to those of standard active learning, particularly in the Bayesian neural network experiment.

---

> ### Author Response · Authors · 2020-11-18
> **Thank you for your review. Part 2.**
>
> > The authors address the above concern by claiming that this paper is only about the "statistical bias" of active learning. I find this unsatisfactory since without the aspect of downstream learning, this is simply an estimation of an expectation with samples from a mismatched distribution.
>
> The concern also seems to stem from a misunderstanding. We have added clarifying remarks to the paper to elaborate the contrast between statistical and overfitting bias.  Existing active learning papers only consider statistical bias without acknowledging that this is not the whole story.  An important contribution of our paper is in pointing this out and analysing more than just the statistical bias empirically in Section 8.
>
> Having the statistical bias equal to zero is what one needs to produce a valid training algorithm, that is one whose parameters will asymptotically converge (with a sufficiently powerful optimizer and sufficient data) to the true optima of the risk.  As such, eliminating statistical bias is key to ensuring models are trained correctly.
>
> In contrast, having the overfitting bias equal to zero would be equivalent to having no generalization gap, i.e. a case where our training empirical risk is an unbiased estimator of the true risk.  Solving the generalization gap entirely is not only beyond the scope of this paper, but completely infeasible; it would be equivalent to having access to the test data during training.  Even outside of an active learning context, researchers have only ever been able to characterize overall bias through loose train-test generalization bounds based on e.g., Rademacher complexity.
>
> Putting this together, we see that characterizing and eliminating the statistical bias is the most we can realistically hope to achieve at present, and we think it is a real strength of our work that we have not only managed to do this, but also accurately assess its limitations.
>
> > The experimental section could have used significant strengthening.
>
> We understand the desire for more experiments, although we note that the main contributions of the paper are theoretical and the paper is already very long including the extensive proofs in the appendices. We have added numerous experiments (see general response to reviewers for a list), including a number of ablations, and hope these go some way to alleviating your concerns.  We also ask you to consider what can reasonably be achieved in a single 8 page paper given the substantial range of contributions that have already been made (e.g. 6 separate non-trivial theorems and associated proofs).

---

### Official Review · AnonReviewer1 · 2020-10-29
**This work analyzes bias-variance of estimators commonly used for active learning. I think the analyses and particularly the connections drawn between overfitting bias vs bias induced by the active learning strategies themselves is a very interesting analysis. Other than a few clarification questions I have, which I outline below, I think this paper is a useful contribution to the ICLR and ML community.**

**Rating:** 7
**Confidence:** 3

**Review:**

The motivation of the paper is well done and it is written clearly as well as well organized although could be improved for the sake of more clarity and precision.

This kind of analysis is important to understand the successes/failures of active learning strategies commonly used in ML and grounding the analysis in bias-variance trade-off is a very useful start.

I have a couple of clarifying questions that I hope the authors can elaborate on. My main suggestion would be to have an even more explicit discussion (in a separate subsection) on connections to the loss + different acquisition/active learning strategies.

In the analysis of Thm 5, I don't understand what the authors mean by "when things are working well", the expected variance should be low for term 2.

The analysis in Section 7 and connections to fitting model parameters vs acquistion strategies is also unclear. What exactly is the choice of R_X when the acquisition strategy minimizes $R_{PURE}$ or $R_{SURE}$? Rather why should optimization itself not use these estimators (maybe some thing else other than $\hat{R}$ to update model parameters?

I think elaborating these will help readers separate the effects of i) acquisition strategies, ii) updating parameters and ii) choice of risk estimators to be used in both cases. All these moving parts are meshed together in between focusing on proofs. I think if the authors could elaborate on how these play together in a more prescriptive manner, it will help the manuscript.

In the discussion, the authors talk about active model selection. What is active model selection and can authors elaborate what they mean the proposed estimators are useful in this setting?

---

> ### Author Response · Authors · 2020-11-18
> **Thank you for your review.**
>
> > main suggestion would be to have an even more explicit discussion (in a separate subsection) on connections to the loss + different acquisition/active learning strategies
>
> > help readers separate the effects of i) acquisition strategies, ii) updating parameters and ii) choice of risk estimators to be used in both cases.
>
> Thank you for your suggestion, we have added a new section (Section 4) to elaborate these points and help separate the issues. We clarify that our approach tries to separate as much as possible:
> i) the choice of acquisition proposal distribution;
> ii) the risk estimator;
> iii) optimization.
>
> > Thm. 5… “when things are working well”
>
> We mean that the acquisition proposal is putting more probability on selecting surprising/informative data. This should result in the weighted loss having lower variance than the unweighted loss, because the two are anti-correlated. We have rewritten this to clarify.
>
> > What exactly is the choice of $R_X$ when the acquisition strategy minimizes $R_{\text{SURE}}$ or $R_{\text{PURE}}$? Rather why should optimization itself not use these estimators...?
>
> I think we previously introduced some confusion in the presentation of what is now Section 8, which the newly uploaded version should clarify. In short, any risk estimator can be used to optimize the model. When we wrote $R_X$ we meant it as a placeholder to stand in for *any* risk estimator, not a specific one (we have now updated the notation for this to hopefully make it clearer). Whatever estimator you have chosen to optimize your parameters using, the expression in equation (16) estimates the bias introduced by overfitting for that *training* estimator.  As such, we are already doing what you suggest in terms of “should optimization itself not use these estimators.”
>
> > What is active model selection and can authors elaborate what they mean the proposed estimators are useful in this setting?
>
> In short, active model selection is the idea we had where one uses active sampling of test data to efficiently estimate the risk, e.g. so that we can choose between models or tune hyperparameters.  Our work is an essential step towards being able to do this as here the focus is on accurate risk estimation (rather than training) and so eliminating the statistical bias is essential (noting that the overfitting bias in the training estimate no longer applies here). We included this as a potentially exciting area of future work, but (as noted by another reviewer) we seem to have slightly overemphasised it and gotten ahead of ourselves a little! For the time being, we will simply point out that after optimization is complete, there is no reason not to use $R_{\text{SURE}}$ rather than $\tilde{R}$ for any risk estimates in typical cases.

---

### Official Review · AnonReviewer2 · 2020-10-31
**Informative Analysis of Bias in Active Learning**

**Rating:** 8
**Confidence:** 4

**Review:**

### Summary

This paper analyzes the bias of models in pool-based active learning settings where the sampling procedure is probabilistic (non-deterministic). It proposes two unbiased estimators of the population risk that weight the loss for each sampled data point. Empirical experiments demonstrate that these unbiased estimators work well for active learning settings with under-parameterized models, but are less effective for over-parameterized models. For the latter case, the authors provide meaningful insights into why biased estimators of population risk may actually be beneficial for over-parameterized models.

### Reasons for score

**Strong points**

1. The authors identify a common problem in active learning, namely the problem of non-IID data. The authors also propose theoretically-sound solutions to addressing the active learning bias.

2. The authors experientally demonstrate their estimators in both linear regression and a deep learning experiment, showing how their estimators are beneficial in one case but not the other.

3. The paper explains why the proposed estimators may have lower variance than the standard estimator for population risk.

**Room for improvement and/or more clarity**

1. The paper mentions the "quality" of the "acquisition proposal" several times (e.g., p.4, p.6, Figure 2, etc.). However, it is unclear what is meant by "quality." What makes for a "good" acquisition proposal?

2. The paper only discusses the setting where the active learning sampling procedure is probabilistic (non-deterministic), with non-zero support over all remaining data points in the pool. However, many active learning procedures proposed in the literature are deterministic (e.g., choosing the K examples with highest loss, or the greedy core-set approach proposed by Sener & Savarese (2018)). Could the authors provide more context for bias in these deterministic active learning strategies?
3. While I did not spend time reading the appendix in detail, it seems that points for the BNN were selected under $\tilde{R}$ (the biased sub-sample empirical risk). How do the results compare if the points were selected under $R_{SURE}$, which is a more relevant setting given that $R_{SURE}$ is the proposed estimator?
4. Other than running active learning twice (once using $\tilde{R}$ and once using $R_{SURE}$), are there methods to determine when one would be preferred over the other, especially in deep learning settings?


### Minor comments /  typos / formatting concerns

(these did not affect my score)

1. Please use vector graphics for figures. For example, save the figures as SVG or PDF files (instead of JPG or PNG) and then include them in the LaTeX file. In particular, the legend in Figure 2a is difficult to read.

2. Figure 2: Does "shading is one standard deviation" mean +/- 1 std (for a total of 2 std), or +/- 0.5 std (for a total of 1 std)? Please clarify.

3. In Equation (5), why not factor out the $1/N$ in front of the summation?

4. The authors write,
> "Using our corrective weights recovers the ideal line." (p.6)

    Unless I am reading the graph wrong, this sentence is slightly misleading. It seems like the linear regression model trained using the corrective weights much more closely approximates the ideal line, but doesn't recover the ideal line exactly.

5. In general, the paper is written in an "opinionated" or imprecise manner which detracts from the main points of the paper. Words like "surprising" and "greatly" are unnecessary. Phrases such as "better variance" should be replaced with more precise terminology, such as "lower variance".

---

> ### Author Response · Authors · 2020-11-18
> **Thank you for your review.**
>
> > “quality” of the “acquisition proposal”
>
> By high-quality proposal, we mean one that will produce a low variance estimator, with the lowest variance proposal being that which is described in Theorem 6.  We have made updates to make this more precise.
>
> > many active learning procedures proposed in the literature are deterministic
>
> This is important and deserves more discussion. We have added elaboration to both a new section 4 and what is now section 8.
>
> First note the deterministic approaches are still biased, they just have zero variance given the data.  Here the bias cannot be eliminated: the empirical risk under a deterministic acquisition scheme is not a random variable given the data. We could write it as $E[R_{\text{det}}]-R$, where $R_{\text{det}}$ is now deterministic given the data. Though this is not very enlightening, it is impossible to say anything more precise without focusing on a particular deterministic acquisition strategy which would be beyond the scope of this paper.
>
> As a further point, it is always possible to adapt a deterministic scheme into a stochastic acquisition proposal. Many commonly used schemes involve an argmax to select the K most informative points at each step. This can be easily replaced with a softmax function, with the temperature adapted to set the desired behaviour. We discuss this approach briefly in the conclusion and experiment description. An alternative method for schemes that do not use an argmax (some k-center coreset approaches perhaps) is an analogy to epsilon-greedy exploration: select a point uniformly at random with some low probability epsilon. We have added further discussion of these issues to section 8 and have added an epsilon-greedy baseline for the linear regression proposal distribution to the Appendix in Figure 6.
>
> > How do the results compare if the points were selected under $R_{\text{SURE}}$?
>
> Our aim was to evaluate the risk estimator directly, separating out the effect that different estimators might have on the data which was selected. But we agree this is a sensible experiment and have added it as Figure 7 in the Appendix. It shows that using $R_{\text{SURE}}$ to train the acquisition models has no significant effect on performance.
>
> > are there methods to determine when one would be preferred over the other?
>
> Though this is difficult to characterise exactly, there are a few things we can say:
> - The more closely the acquisition proposal distribution approaches the optimal distribution, the relatively better our estimators will be.
> - The less overfitting we anticipate, the more useful our estimators will typically be.
> - $R_{\text{SURE}}$ and $R_{\text{PURE}}$ will tend to have more of an effect for highly imbalanced datasets, as the biased estimator will over-represent actively selected but unlikely datapoints.
> - The less the training data reflects the true data distribution, the less useful our estimators are is (since this makes it impossible to estimate the true risk in any case).
>
> We have elaborated on this in the Conclusions.
>
> We also note that after optimization is over which should generally prefer $R_{\text{SURE}}$ (other than the unexpected hypothetical scenario where it is much higher variance than $\tilde{R}$).
>
> > vector graphics
>
> Thank you for pointing this out, we have replaced the figures with pdfs.
>
> > Figure 2... standard deviation
>
> The shading is +/- 1 std. We have clarified this in the paper.
>
> > Equation (5)... factor out the 1/N?
>
> We left it in the summation as it forms parts of the weight definition, but you are right that it can be factored out if desired.
>
> > recovers the ideal line
>
> Yes you are correct, our wording was imprecise here and have corrected this.
>
> > “opinionated” language
>
> Thank you for this comment, we have made edits to remove these.

---

### Author Response · Authors · 2020-11-18
**Thank you all for your reviews**

We want to thank all the reviewers for their efforts and the thoughtful and attentive examination of our work that they have offered.

We were glad to see that reviewers suggested that the paper offered “important” and “informative analysis” [AR1 and AR2] of an “interesting problem” [AR4] with “meaningful insights” and “theoretically-sound solutions” [AR2] and that it was “presented very clearly” [AR3]. They further provided excellent suggestions on how to improve the paper further, which we discuss in comments below and have integrated into the newly uploaded draft. In addition to individual points, there were a couple shared themes.

AR1 and AR3 asked for more clarity on the interaction between acquisition proposal distributions, risk estimators, and optimization. We have added a new section (Section 4), as suggested, which elaborates on these points.

A number of the reviewers also asked for further experimental support. As such, we have now added the following extensive range of new experiments, several of which were suggested by the reviewers, with a focus on showing that our results are not sensitive to specific modelling assumptions:
- Acquiring points using $R_{\text{SURE}}$ rather than $\tilde{R}$. [Figure 7] [AR2]
- Using a multi-layer perceptron rather than a convolutional Bayesian neural network. [Figures 10d and 10e] [AR4]
- Using balanced MNIST training data rather than unbalanced. [Figures 3f and 10c] [AR4]
- Using Monte Carlo dropout rather than Radial BNN. [Figures 3e and 10b]
- Using FashionMNIST in addition to MNIST. [Figures 2c, 3d, 4c, and 10a] [AR4]
- Using a wider range of temperatures for the acquisition proposal distribution. [Figures 9a-f]
- Using a different acquisition proposal distribution for the linear regression experiment. [Figures 6a and 6b] [AR2,4]

We further emphasise that our core contributions are theoretical rather than empirical and note that we offer extensive results on this front (including six Theorems with non-trivial proofs and a novel framing of bias in active learning).

---

### Decision · Program_Chairs · 2021-01-07
**Final Decision**

**Decision:**

Accept (Spotlight)

**Comment:**

This is an excellent paper that provides analytical and empirical insights on the sample selection bias of pool-based active learning. It provides two very practical methods of removing the bias.  Also, it shows that in over-parameterized networks (like modern neural networks), the active learning bias could actually be useful.   I enjoyed reading the paper.  Reviewers are mostly very positive about the paper.  Experiments in the initial version were limited, and the authors have since added more experiments.   With the increasing interest on learning with limited data, this paper is very timely and useful.  I expect the paper to be of interest to many in the community.